



# The added value and potential of long-term radio occultation data for climatological wind field monitoring

Irena Nimac[1], Julia Danzer[1], Gottfried Kirchengast[1,2]

[1]Wegener Center for Climate and Global Change, University of Graz, Graz, 8010, Austria

[2]Institute of Physics, University of Graz, Graz, 8010, Austria

*Correspondence to*: Irena Nimac ([irena.nimac@uni-graz.at](mailto:irena.nimac@uni-graz.at)) and Julia Danzer ([julia.danzer@uni-graz.at](mailto:julia.danzer@uni-graz.at))

**Abstract.** Global long-term stable 3D wind fields are a valuable information for climate analyses of atmospheric dynamics. Their monitoring remains a challenging task, given shortcomings of available observations. One promising option for progress is the use of radio occultation (RO) satellite data, which enable to derive wind fields based on the geostrophic and

gradient wind approximations. In this study we focus on three main goals, explored through European Re-Analysis ERA5 and RO datasets, using monthly-mean January and July data over 2007–2020 with a 2.5° × 2.5° resolution. First, by comparing ERA5-derived geostrophic and gradient wind speeds to the ERA5 original wind speed, we examine the regions of validity for both these approximations. Second, to assess the potential added value of RO-derived geostrophic and gradient winds, we test how well they agree with the corresponding ERA5-derived winds. Third, we evaluate the potential of the RO

wind fields relative to the ERA5 original wind fields. With this three-step analysis we decompose the total wind speed bias into a bias resulting from the approximation and the systematic difference between the RO and ERA5 datasets. We find that the geostrophic approximation is a valid method to be used to estimate tropospheric winds, while the gradient wind approximation works better in the stratosphere. Both approximations generally work well in the corresponding altitude regions, within 2 m s$^{-1}$ accuracy almost globally (latitudes 5°–82.5°), with some exceptions in the winter hemisphere:

monsoonal area at the lower altitudes, northern polar regions at higher altitudes, and larger mountain regions throughout all investigated altitude levels. RO- and ERA5-derived geostrophic and gradient winds mostly showed very good agreement, generally within 2 m s$^{-1}$. However, when studying the decadal trend, temporal change in the systematic differences higher than 0.5 m s$^{-1}$ per decade was found. This points to a potential effect of observing system changes in ERA5 around the year 2016. The overall high accuracy of the monthly-mean wind fields, backed by the long-term stability of the underlying RO

data, highlights the added value and potential benefit of RO-derived winds for climate monitoring and analyses.



## 1 Introduction

Wind field measurements have an important role in numerical weather prediction (NWP) and in atmospheric sciences for
understanding climate dynamics and chemistry. As they serve as initial conditions in NWP models, their accuracy is of great
importance. Besides, such data are also regularly assimilated in reanalysis systems, contributing to advances in climate
science (Stoffelen et al., 2005; Eyre et al., 2020). Even though nowadays there is an increased portion of different techniques
for measuring wind speeds, having accurate global 3D wind information is still a demanding task due to certain limitations
of specific observation techniques (Stoffelen et al., 2005, 2020). While some techniques have generally good spatial
coverage (e.g., meteorological stations, ships, buoys, scatterometer winds from satellite radars), they only provide wind
information on single levels, lacking the vertical wind profile. On the other hand, techniques providing vertical profiles (e.g.,
wind profilers, radio-sounding data, pilot balloon data) have relatively coarse spatial coverage. Hence over larger parts in the
southern hemisphere, such as oceans, both fine horizontal and vertical wind information is a problem (Stoffelen et al., 2005,
2020).

Altitude-resolving satellite data can help overcome these problems between profiling information and good global coverage.
European Space Agency's (ESA) Earth Explorer mission Aeolus utilizes the active Doppler Wind Lidar method to measure
wind from surface up to 30 km altitude (Stoffelen et al. 2005; Kanitz et al. 2019). The assimilation of this dataset resulted in
improvement in NWP forecasts (Rennie et al., 2021; Žagar et al., 2021), as well as helped to better understand and analyse
atmospheric dynamics such as Kelvin waves (Žagar et al., 2021) or gravity waves (Banyard et al., 2021). However, due to its
quite short time period (launched in August 2018), these data are not suitable for climate change analyses. Another technique
to derive vertical profiles is Global Navigation Satellite System (GNSS) radio occultation (RO), where the thermodynamic
state of the atmosphere is obtained based on the transmitted GNSS radio signals refracted by the Earth's atmosphere
(Kursinski et al., 1997; Steiner et al., 2011; Mannucci et al., 2020). The advantage of RO is its unique combination of global
coverage, high vertical resolution, high accuracy, long-term stability, and multi-mission data consistency (e.g., Anthes, 2011;
Foelsche et al., 2011; Angerer et al., 2017; Zeng et al., 2019; Steiner et al., 2020a). The RO data sets are assimilated into
operational weather forecasts (e.g., Healy and Thépaut, 2006; Buontempo et al., 2008; Cardinali 2009) and long-term
reanalyses (e.g., Hersbach et al., 2020; Kobayashi et al., 2015; Gelaro et al., 2017), and are used in climate analysis studies
(e.g., Steiner et al., 2011, 2020b; Stocker et al., 2021).

While RO does not directly provide wind information, winds can be estimated from geopotential information using the
conventional geostrophic and gradient wind approximations. The geostrophic approximation is a commonly used wind
method in diagnostic studies. Although its utility in the troposphere was found to be good, in the winter extratropical
stratosphere significant overestimation of the polar jet stream (~10-20%) is found due to the neglect of local curvature





effects (e.g., Boville 1987; Elson 1986; Randel 1987). Boville (1987) comments that the error in the meridional wind component is comparable to the error in the zonal wind component at all levels. To overcome this problem, one can use the

gradient wind approximation which involves an additional centrifugal term on top of the geostrophic balance. This method generally gives better results for stratospheric winds (e.g., Scherllin-Pirscher et al., 2014), however during intense wave activity, it produces large errors in high-latitude stratospheric regions (Elson 1986; Randel 1987).

Besides a seasonal and altitudinal dependence of the validity of the geostrophic and gradient wind approximation, another limitation is its breakdown towards the equatorial region as the Coriolis parameter approaches zero. Oberheide et al. (2002)

linearly interpolated geostrophic wind fields between ±10° latitude, Elson (1986) started with 4° N as the lowest latitude, Randel (1987) and Boville (1987) start at 10° N, while Scherllin-Pirscher et al. (2014) and Verkhoglyadova et al. (2014) left out the regions of ±15° and ±10°, respectively, from their wind estimation studies.

Even though the mentioned wind approximations are well demonstrated methods to derive dynamics from satellite information based on the mass (geopotential height) field (e.g., Oberheide et al., 2002; Scherllin-Pirscher et al., 2014, 2017;

Verkhoglyadova et al., 2014), their accuracy and validity for different latitudinal and altitudinal regions, as well as the regions of breaking-down, have not been thoroughly investigated. Several validation studies were made few decades ago using measurements such as rawinsonde (e.g., Wu and Jehn 1972) or climate models (e.g., Boville 1987; Randel 1987).

To our knowledge, there are no up-to-date studies dealing with a rigorous evaluation of the geostrophic and gradient wind approximations with a clear focus on climatological long-term wind field monitoring. This is especially important in regard

to recent improvements in both measurements and climate models in terms of temporal and spatial resolution as well as the parametrizations and processes included (Rummukainen, 2010). While Elson (1986) compared the estimated geostrophic wind with the one derived using higher-order approximations which accounted for horizontal wave flux convergence terms, Boville (1987) points out that such an approach does not give any information of how close the higher-order approximation is to the real wind. Hence, to test the quality of the used wind approximations, one needs to have a dataset which contains

information on both the pressure-geopotential height relation (thermodynamics) and the real wind (dynamics), such as climate model or reanalyses.

Hence, in this study we focus on three main goals. First, we use ERA5 reanalysis data to evaluate the geostrophic and gradient-wind approximations for representing monthly-mean wind fields down to synoptic-scale spatial resolution (about 300 km, 2.5° × 2.5° grid), as a method potentially vitally helpful for deriving long-term climatic changes in dynamics (wind

fields) from long-term thermodynamic satellite data (mass fields). Here we in particular aim to understand the impact of the missing ageostrophic contribution and centrifugal term in the approximation, to better understand its quality for deriving such monthly-mean winds. We examine both horizontal and vertical regions in order to assess where selected approximations perform better. Second, we evaluate the utility of RO-derived monthly-mean winds, for their potential added value as a separate wind field monitoring climate data record providing improved long-term stability. This is made by



comparing RO-derived winds with corresponding ones estimated based on the ERA5 data. Lastly, we show the potential of RO-estimated winds in describing original ERA5 wind fields. The study builds upon and substantially advances a preliminary study by Nimac et al. (2023).

The paper is structured as follows: In Sect. 2 we describe the data and the method used in the study. The results are presented in Sect. 3, while Sect. 4 covers the discussion part. Conclusions and perspectives are finally given in Sect. 5.

## 2 Data and study method

In this analysis we used global monthly-mean ERA5 reanalysis (Hersbach et al., 2020) and multi-satellite RO OPSv5.6 data (Angerer at al., 2017; Steiner et al., 2020a) during the joint time period 2007 to 2020. We analysed the global wind data at 2.5° latitude × 2.5° longitude resolution, in the altitude region from near surface (1000 hPa) up to the middle stratosphere (10 hPa, about 32 km). We chose January and July as two representative months for the winter and summer season. A further advantage of those two months is that the strongest wind speeds in jet stream regions are observed (e.g., Scherllin-Pirscher et al., 2014, 2017), reaffirming that these serve as an adequate test data for the goals of this study. For both months, we calculated the long-term monthly-mean wind speed fields over the 14-year period of 2007 to 2020.

We are aware that the ERA5 reanalysis also includes RO information, through its data assimilation process that ingested these and many other observation types, and hence depends somewhat also on RO data. However, all major state-of-the-art (re)analyses do assimilate RO data in our time range of interest since 2006 (start of the "U.S. COSMIC" and "European Metop" RO multi-satellite era). Having an overall suitable and high-quality reference dataset, which does not assimilate RO data is hence essentially not feasible. For example, both JRA-55c and MERRA reanalyses do not assimilate RO data, while the recent versions JRA-55 and MERRA-2 do so. However, there are also additional differences, such as JRA-55c does not assimilate any satellite data, which lowers the data quality in the upper troposphere and lower stratosphere drastically (Kawatani et al., 2020). Similarly, MERRA is based on an older model system, in quality inferior and not comparable to ERA5 or MERRA-2. Hence the accuracy of such non-RO reanalyses is not suitable to be used as a reference for original winds in this study. Therefore, and since we know from other studies (e.g., Scherllin-Pirscher et al., 2017; von Schuckmann et al, 2023, Sect. 3 therein) that the specific choice of (re)analysis plays no significant role for such an evaluation study design, we find ERA5 the most suitable choice.

## 2.1 ERA5 reanalysis data

As a state-of-the art reference dataset to test the validity of the geostrophic approximation, we used the European Re-Analysis 5th Version (ERA5) of the European Centre for Medium-Range Weather Forecasts (ECMWF). On the chosen 2.5° latitude × 2.5° longitude grid of the selected monthly-mean data, we extracted eastward-wind and northward-wind





components, for computing the original wind speeds, as well as isobaric geopotential height data (geopotential fields on
pressure levels), for deriving the geostrophic winds and, on top of them, gradient winds. We term the ERA5 original wind
speeds as $ERA_{orig}$, ERA5 geostrophic ones as $ERA_{geos}$, and ERA5 gradient winds as $ERA_{grad}$.

## 2.2 RO satellite data

The RO multi-satellite climatologies are derived from the satellite missions CHAMP (Wickert et al., 2001), C/NOFS (de la
Beaujardiere et al., 2004), F3C (Anthes et al., 2008), GRACE (Beyerle et al., 2005; Wickert et al., 2005), MetOp (Luntama
et al., 2008), and SAC-C (Hajj et al., 2004). Phase data were derived at UCAR/CDAAC (University Corporation for
Atmospheric Research/COSMIC Data Analysis and Archive Center), and further processed at the Wegener Center (WEGC)
using the Occultation Processing System OPSv5.6 (Angerer et al., 2017; Steiner et al., 2020a). Based on the atmospheric
bending of the GNSS signals during the occultation sounding, it is possible to retrieve atmospheric refractivity profiles.
From these, air density and pressure profiles as a function of altitude, or geopotential height, can be accurately derived based
on the refractivity equation, the equation of state, and the downward integration of the hydrostatic equation. In this way,
geopotential height profiles as a function of pressure levels can be obtained with unique accuracy and form the basis for the
wind field derivation (for a more detailed description see Scherllin-Pirscher et al., 2017).
Monthly 2.5° × 2.5° gridded data were then derived using aggregated atmospheric profile data weighted according to the
longitude-latitude distances of each profile to the bin centre using Gaussian-radius-lon-lat weighting within a radius of 600
km. This corresponds to using equal-area cell size around each grid-cell center location; hence the 2.5° x 2.5° sampling grid
does not imply that we get smaller cell areas towards the poles as a result of meridian convergence. On average, the number
of RO profiles is around 60 000 profiles per month. To derive the RO geostrophic wind speeds, we used isobaric
geopotential height data (as for ERA5), while gradient wind fields are further estimated based on the derived geostrophic
wind. We term these RO-derived wind speeds as $RO_{geos}$ for geostrophic RO wind and $RO_{grad}$ for RO gradient wind.
RO data show high accuracy and vertical resolution of the relevant isobaric geopotential height fields over the altitude region
of 5 km to 35 km (Scherllin-Pirscher et al., 2017; Steiner et al., 2020a). In the moist lower to middle troposphere region,
background information of (re)analysis data supports the RO thermodynamic data retrieval from atmospheric refractivity
(Scherllin-Pirscher et al., 2017; Li et al., 2019). Towards higher altitudes into the upper stratosphere, the impact of residual
errors due to measurement noise and ionosphere starts to increase (e.g., Danzer et al., 2013, 2018; Liu et al., 2018),
decreasing the accuracy of the RO-retrieved isobaric geopotential height data. Hence, we focused our evaluation of RO
utility up to the middle stratosphere (about 35 km, 10 hPa level).



### 2.3 Study method

We studied the regions of validity of the geostrophic and gradient wind approximation (first goal) as the difference of
ERA$_{geos}$ and ERA$_{orig}$ wind fields and of ERA$_{grad}$ and ERA$_{orig}$ wind fields, respectively. This approach allows to study solely
the bias resulting from the approximations. Further, we evaluated the differences of the RO-derived geostrophic and gradient
winds from the reanalysis-derived ones (second goal) in terms of the RO$_{geos}$ vs. ERA$_{geos}$ and RO$_{grad}$ vs. ERA$_{grad}$ difference
(Fig. 1). By using this two-steps evaluation method, we first quantitatively test the adequacy and quality of the selected wind
approximation methods based on the reanalysis data, while in the second step we estimate the systematic difference between
RO and the reanalysis data for the wind derivation, which basically relates to a bias between the two datasets. This way we
show the contribution of each component to the total difference between RO-derived wind field and ERA5 original winds.

For inspecting horizontal latitude-longitude maps, we concentrated on the four representative levels 200 hPa, 150 hPa, 50
hPa and 10 hPa, which represent the higher troposphere/tropopause, lower stratosphere and middle stratosphere regions,
respectively. As a focus region, we examined latitudinal-altitudinal cross-sections of the respective wind speed differences
averaged over 140°E–160°E longitudinal area. This longitudinal region is selected since the observed larger differences were
mainly found there (i.e., roughly jet-stream core region). To assess the added value of RO data compared to ERA5 in terms
of their temporal homogeneity and long-term stability, we analyse temporal differences in wind derived from two datasets.

To derive wind fields, one commonly starts with the equations of zonal and meridional momentum (Holton, 2012).
However, due to the complexity of solving these non-linear partial differential equations, some assumptions and
simplifications are useful to estimate approximate wind components. To derive them in line with the focus of this study from
thermodynamic data (such as provided by RO) we first use the geostrophic approximation. In this approximation, most of the
horizontal momentum equation terms are neglected, except for the Coriolis force term, which is balanced by the pressure
gradient force. In the isobaric coordinate system and applying the so-called local-$f$-plane formulation, zonal ($u_{geos}$) and
meridional ($v_{geos}$) geostrophic wind components are given by the relations (Holton, 2012; Scherllin-Pirscher et al., 2014;
Verkhoglyadova et al., 2014):

$$u_{geos} = \frac{-1}{f(\varphi)\,a} \frac{\partial \Phi}{\partial \varphi} \tag{1}$$

$$v_{geos} = \frac{1}{f(\varphi)\,a\cos(\varphi)} \frac{\partial \Phi}{\partial \lambda} \tag{2}$$

where $f(\varphi)$ is the local Coriolis parameter, $f(\varphi) = 2\Omega\sin\varphi$, with $\Omega = 7.2921\times10^{-5}$, a is the Earth's radius, $\Phi$ denotes
geopotential on isobaric levels, $\varphi$ is geographic latitude, and $\lambda$ longitude. Geopotential $\Phi$ is calculated as $\Phi = Z\,g_0$ where $Z$ is
geopotential height and $g_0 = 9.80665$ m s$^{-2}$ the standard gravity constant. Hence, to derive geostrophic wind fields, we need



geopotential height fields at pressure levels as information. As shown in Scherllin-Pirscher et al. (2017), alternatively the geostrophic wind could also be derived as the gradient vector of the Montgomery potential at potential temperature surfaces, but the results do not differ from the geopotential-based derivation in the local-$f$-plane formulation used here.

Based on the estimated geostrophic wind speeds, the gradient wind approximation is used on top. In this approximation, the pressure gradient term is balanced not only by the Coriolis force but by the Coriolis and the centrifugal force together. The equations for zonal ($u_{grad}$) and meridional ($v_{grad}$) gradient wind components are hence functions involving the geostrophic wind components as their backbone, calculated as (Holton, 2012; Scherllin-Pirscher et al., 2014):

$$u_{grad} = F(u_{geos}, \varphi) = \frac{-f(\varphi) \pm \sqrt{f^2(\varphi) + 4 f(\varphi) u_{geos} \tan(\varphi)/a}}{2 \tan(\varphi)/a} \tag{3}$$

$$v_{grad} = F(v_{geos}, u_{grad}, \varphi) = \frac{v_{geos} f(\varphi)}{f(\varphi) + u_{grad} \tan(\varphi)/a} \tag{4}$$

Here, the +/- sign refers to Northern (+) resp. Southern Hemisphere (–). The total wind speed is calculated as a squared root of the sum of the squared zonal and meridional wind components, and we term this $V_{orig}$ for the original wind, $V_{geos}$ for the geostrophic wind and $V_{grad}$ for the gradient wind.

This procedure was applied to both ERA5 and RO geopotential fields to estimate corresponding geostrophic winds and, on top, gradient winds. To assess the added value of the gradient wind approximation on top of the geostrophic approximation, we estimate the gradient wind delta-difference field $\Delta V$ defined as:

$$\Delta V = |Bias_{geos}| - |Bias_{grad}| = |V_{geos} - V_{orig}| - |V_{grad} - V_{orig}| \, , \tag{5}$$

$$\Delta V > 0 \rightarrow |Bias_{grad}| < |Bias_{geos}| \, ,$$

$$\Delta V < 0 \rightarrow |Bias_{geos}| < |Bias_{grad}| \, .$$

In using this convenient absolute delta-difference metric for inspecting the additional bias reduction or bias increase by the gradient wind approximation vs. the geostrophic approximation, the regions where both approximations give similar values will be suppressed (delta-difference near zero), while areas with larger delta-difference will stand out. Positive delta-difference values indicate better estimation of the original wind by the gradient wind approximation, while negative delta-difference values represent the opposite – better representation of the original wind by the geostrophic approximation.

Since deriving geostrophic wind fields is based on the horizontal derivatives of geopotential height, prior horizontal smoothing of RO geopotential field was needed (Elson 1986). Here, we first smoothed the 2.5° × 2.5° geopotential fields using a 5-point Gaussian filter in both longitudinal and latitudinal direction. In the latitudinal direction, the last two latitude circle grid lines were excluded from the analysis, since they are needed as filter margin. Additionally, one more grid line (85°) was discarded after calculating the derivative according to Eq. (1). The final latitudinal range used for the RO-derived





fields is ±82.5°. Related to this, due to the lower number of soundings over the polar caps (as well as the complexity of
calculations over polar regions), it is justified to exclude these few polar latitude circles from the analysis in both RO and
ERA5 wind fields.

As the Coriolis parameter $f(\varphi)$ approaches zero near the equator, the approximations are not valid in those areas. A separate
wind analysis based on the thermal wind balance was done by Danzer et al. (2023) for the equatorial region. Still, as in our
data grids the lowest latitude bin grid lines are at ±1.25° latitude, it was possible to calculate winds for all climatological
bins, though values close to the equator lose physical meaning. This way we determined the region of approximation break-
down by comparing the approximation bias to some commonly used accuracy requirement values.

We used the monthly-mean geopotential data at isobaric levels for the January and July months in the period 2007–2020 to
derive the geostrophic wind components using equations (1) and (2), and gradient wind by equations (3) and (4), and
subsequently computed the speed as the magnitude of the corresponding wind vector. The wind speeds for ERA$_{orig}$, ERA$_{geos}$,
RO$_{geos}$, ERA$_{grad}$ and RO$_{grad}$ were then used to perform our evaluations according to Fig. 1. All calculations, statistical analysis
and visualization were performed using Python programming language and mainly its packages *numpy, xarray,
pymannkendall* and *matplotlib*.

To put the results into context with reasonable wind accuracy requirements (e.g., Stoffelen et al., 2020), we used absolute
requirement values for domains with small wind speeds, while for large wind speeds relative requirement values appear a
more appropriate choice. We chose a difference of ±2 m s$^{-1}$ or a relative difference of ±5 % as requirements, which are
values consistent with the wind observation accuracy target requirements specified by the World Meteorological
Organization (WMO) for various applications, including NWP and climate (Stoffelen et al., 2020; WMO-OSCAR, 2022;
Table 1). Simple linear regression was used to test the long-term temporal stability of the derived wind speed fields. We
estimated the decadal trend rate in difference between RO$_{geos}$ and ERA$_{geos}$ and evaluated this against the WMO-GCOS (2016)
wind measurement stability target requirement of ±0.5 m s$^{-1}$ per decade (see also Table 1).





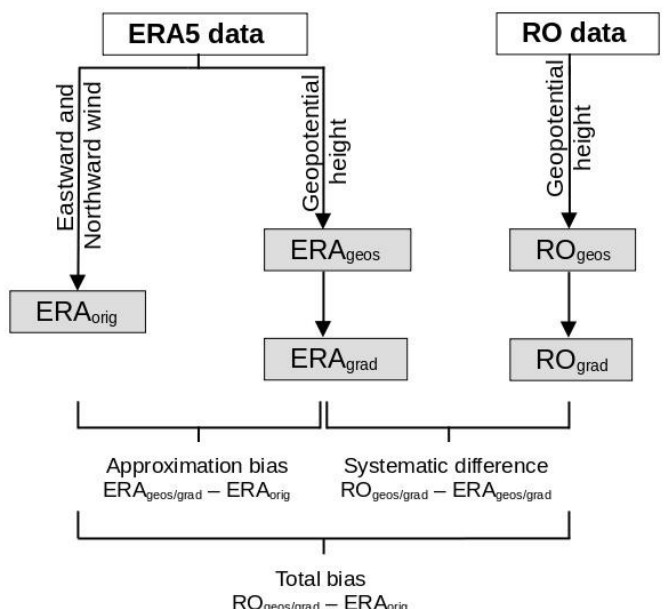

**Figure 1:** Schematic diagram of the three-steps evaluation method. The original ERA5 (ERA$_{orig}$) wind speed is calculated

based on the northward and eastward wind components. In the first step, geostrophic ERA5 (ERA$_{geos}$) and geostrophic RO (RO$_{geos}$) winds are estimated from the corresponding geopotential height data. In the second step, ERA5 gradient (ERA$_{grad}$) and RO gradient (RO$_{grad}$) winds are calculated using previously derived ERA$_{geos}$ and RO$_{geos}$, respectively. Approximation bias is computed as a difference between the estimated ERA$_{geos}$ and ERA$_{grad}$ relative to ERA$_{orig}$, while the datasets' systematic difference is computed as the difference in corresponding wind fields, ERA$_{geos}$ and RO$_{geos}$ or ERA$_{grad}$ and RO$_{grad}$, respectively.

Total bias is computed as the difference between estimated RO winds and ERA$_{orig}$.

**Table 1:** Selected absolute and relative wind speed accuracy requirements used in the study, informed by WMO-GCOS (2016).

| Accuracy Specifications | Absolute | Relative |
|---|:---:|:---:|
| Evaluation of wind approximations | $\pm 2$ m s$^{-1}$ | $\pm 5$ % |
| Temporal stability check | $\pm 0.5$ m s$^{-1}$ per decade | - |




## 3 Results

### 3.1 Approximation bias – ERA$_{geos}$ and ERA$_{grad}$ vs. ERA$_{orig}$

To test the advantages and disadvantages of geostrophic and gradient wind approximation, we compare estimated ERA$_{geos}$ and ERA$_{grad}$ to the original ERA$_{orig}$ winds for January (Fig. 2) and July (Fig. 3). We show the geostrophic wind

approximation bias as a reference bias field, while for gradient wind we present delta-differences calculated based on the equation (5). This delta-difference approach allows us to show only regions where gradient wind approximation estimates original wind notably better (i.e., positive delta-difference values) or notably worse (i.e., negative delta-difference values) compared to the geostrophic balance. Hence, where the delta-difference is small, the approximations give relatively similar wind estimations.

For both approximations some same deviations from the original wind are observed in the regions where the centrifugal term does not contribute much (i.e., in equatorial and tropical regions). At all four selected levels and both seasons, the largest amplitude of the differences is found around the equator from -5° to 5° latitude, as a result of the Coriolis parameter approaching zero. These differences are clearly larger than the selected accuracy threshold of 2 m s$^{-1}$, indicating the region of approximation break down. Additionally, larger oscillatory features (also well exceeding 2 m s$^{-1}$ threshold) are present above

large mountains and obstacles, e.g., in January above Greenland and Himalaya (Fig. 2) and in July above Himalaya and Andes (Fig. 3). Above Himalaya, the amplitude of this pattern is stronger in January when wind speeds over this area are stronger compared to the July ones. In January, wave-like pattern above Himalaya is slightly weaker in amplitude for the gradient wind than the geostrophic, which is not the case for differences above Andes in July. This is because the contribution of the centrifugal term is smaller in tropical areas, but also due to lower wind speeds there. This mountain effect

is noticeable as high up as the mid-stratospheric 10 hPa level (Fig. 2g–h, Fig. 3g–h). At lower levels, larger underestimation of original wind is observed in monsoonal regions over Indonesia in January and tropical Indian Ocean (southward from 5° S) in July (Fig. 2a-d, Fig. 3a-d).

The largest differences between the geostrophic and the gradient wind approximations are present at the lowest 200 hPa (Fig. 2a-b, Fig. 3a-b) and at the highest 10 hPa (Fig. 2g-h, Fig. 3g-h) observed levels. The geostrophic balance describes the

original wind better at the lower levels, especially over the Pacific Ocean in the regions of the sub-tropical jet stream core where gradient wind approximation shows underestimations (~ ±30° lat) (Fig. 2a and Fig. 3a). On the other hand, the gradient wind performs better in the higher altitude regions, specifically for depicting the polar stratospheric jet-stream (~ ±60° lat) (Fig. 2h and Fig. 3h). The gradient wind balance works better for July stratospheric winds, unlike in January when it significantly underestimates wind in the northern polar regions.


**Figure 2:** Long-term (2007–2020) mean approximation bias. Wind speed differences (m s$^{-1}$) between the geostrophic ERA5 (ERA$_{geos}$) and the original ERA5 (ERA$_{orig}$) wind (left column), and the gradient wind (ERA$_{grad}$) delta-difference calculated using equation (5) (right column), at the 200 hPa (first row), 150 hPa (second row), 50 hPa (third row) and 10 hPa (last row)





level for January. Dashed green vertical lines denote the 140°–160° E area for which vertical cross-section is given. Dashed black horizontal lines delineate the ±5° latitude band around the equator and ±82.5° regions toward poles.



**Figure 3:** The same as in Fig. 2 but for July.



To better understand altitudinal-latitudinal behaviour of both approximation biases, we examine its latitudinal vertical cross-section averaged over the region 140°–160° E (Fig. 4). This longitudinal region is selected because both larger deviations from sub-tropical and polar jet-stream are found here (i.e., region of strengths and weaknesses of geostrophic and gradient

wind approximation). In both seasons, both approximations are not valid in the equatorial region between ±5° latitude. Larger differences are also found at the lowest (boundary layer) levels in the NH mid-latitudes, as well as over the Antarctic. Generally, the geostrophic approximation is better in describing the dynamics of the sub-tropical jet stream, while the gradient wind approximation works better at higher levels since the delta-difference is there positive. An underestimation of the stratospheric polar jet stream by the gradient wind approximation is noticeable through negative delta-difference values

in the high-latitude regions (Fig. 4b). Except for the mentioned larger deviations in describing sub-tropical or polar jet-stream, wind speed differences are well within the accuracy requirement of 2 m s$^{-1}$.

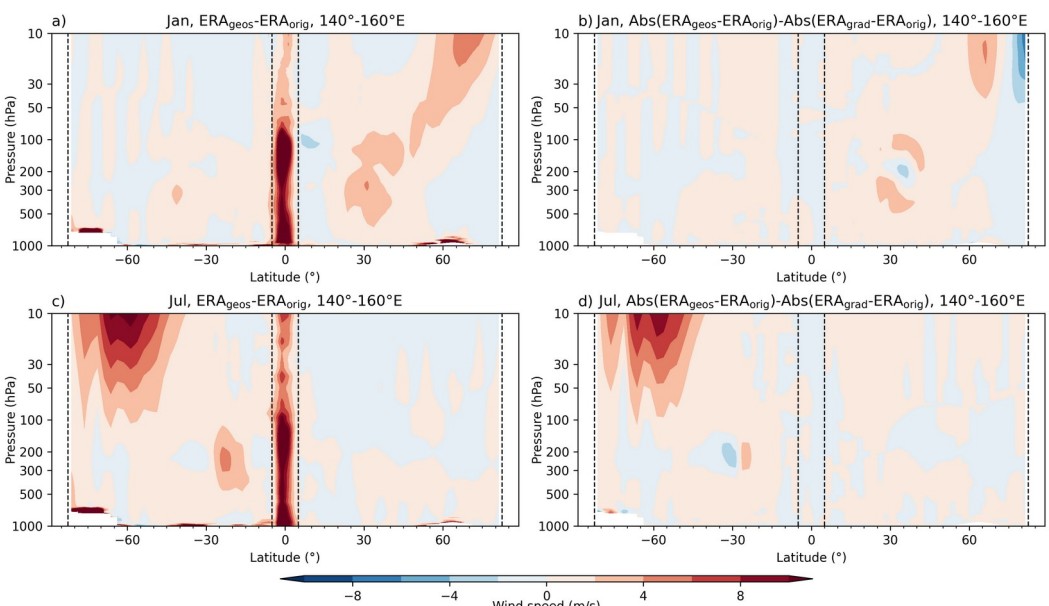

**Figure 4:** Long-term (2007–2020) mean vertical cross-section of the approximation bias. Wind speed differences (m s$^{-1}$)

between the geostrophic ERA5 (ERA$_{geos}$) and the original ERA5 (ERA$_{orig}$) wind (left column) and the gradient wind (ERA$_{grad}$) delta-difference calculated using equation (5) (right column), averaged over the 140°–160° E area for January (top) and July (bottom). Dashed black vertical lines delineate the ±5° latitude band around the equator and ±82.5° regions toward poles.





## 3.2 Systematic difference – RO$_{geos/grad}$ vs. ERA$_{geos/grad}$

In line with the second study goal, we tested how well RO-derived and estimated ERA5 wind fields agree, applying for both data products the same approximation. Here, the equatorial band within ±5° is already excluded from further inspections, building on the results from Sect. 3.1. We focus on understanding the consistency between RO and ERA5 estimated wind speed in those (still) nearly global domains, where the approximations are found to perform well. Since this systematic difference between the two datasets (RO vs. ERA5) is equal for geostrophic and gradient wind, for convenience we show the results only for geostrophic wind.

An inspection of the differences in horizontal maps at selected levels reveals that the overall patterns of the subtropical and polar jet streams are well represented by the RO wind data, but oscillatory patterns above the large mountain ranges are still present (Fig. 5). Generally, the differences are larger over the winter hemisphere, especially in the region of subtropical jet stream at 200 hPa where RO$_{geos}$ over-/underestimates ERA$_{geos}$ around ±20/30° latitude, respectively (Fig. 5a, b). Such pattern might indicate that, based on the geopotential data, the position of jet stream is slightly moved equator-ward in RO compared to ERA5 dataset.

This difference is still present at 150 hPa level, but with lower amplitudes, as well as wave-like features (Fig. 5c, d). At 50 hPa, differences larger than ±2 m s$^{-1}$ are found only in the wave-pattern structures in both seasons above the Greenland, Himalaya and Andes regions (Fig. 5e, f). In January, at the highest level of 10 hPa, besides larger differences above the mentioned mountains, differences up to -4 m s$^{-1}$ are found at high latitudes, corresponding to the position of polar jet stream (Fig. 5g). In July, such deviations from the polar jet-stream are less expressed (Fig. 5h).

Investigating the latitudinal cross-section confirms that the observed larger differences (up to around 6 m s$^{-1}$) in the winter hemisphere correspond to the locations of the sub-tropical jet-stream (Fig. 6). Maximum January amplitude of positive difference at ~300 hPa and negative at ~200 hPa might also point to a somewhat lower location of the jet-stream core in RO compared to the ERA5 data (Fig. 6a).

Wind speed underestimation larger than 10 m s$^{-1}$ is found in both seasons at the lowest levels above the Antarctic. A slight underestimation (smaller than -4 m s$^{-1}$) of the RO polar jet-stream winds is also present in January (Fig. 6a).



**Figure 5:** Long-term (2007–2020) mean systematic difference between RO and ERA5. Wind speed differences (m s$^{-1}$) between the geostrophic RO (RO$_{geos}$) and the geostrophic ERA5 (ERA$_{geos}$) wind, at 200 hPa (first row), 150 hPa (second row), 50 hPa (third row) and 10 hPa (last row), for January (left column) and July (right column). Dashed green vertical lines





denote the 140°–160° E area for which vertical cross-section is given. Dashed black vertical lines delineate the ±5° latitude
band around the equator and ±82.5° regions toward poles.

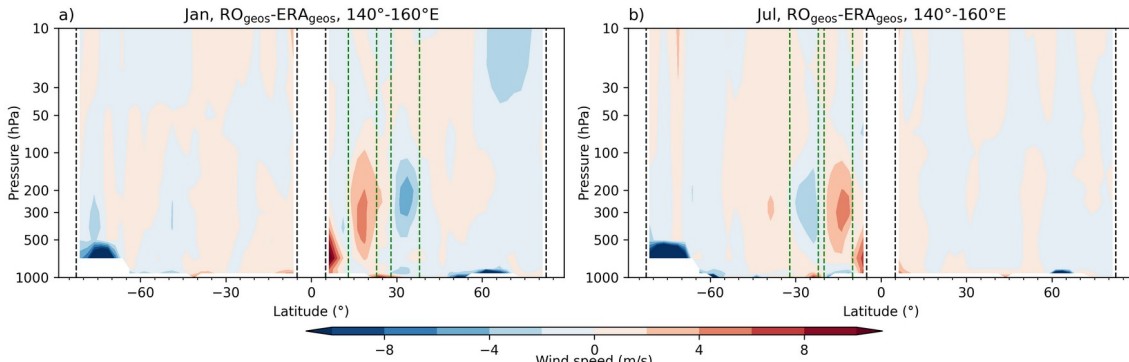

**Figure 6:** Long-term (2007–2020) mean vertical cross-section of the systematic difference between RO and ERA5. Wind
speed differences (m s$^{-1}$) between the geostrophic RO (RO$_{geos}$) and the geostrophic ERA5 (ERA$_{geos}$) wind, averaged over the
140°–160° E area for January (left) and July (right). Dashed green vertical lines denote latitudinal belts used in the further
analysis of long-term temporal consistency of the datasets. Dashed black vertical lines delineate the ±5° latitude band around
the equator and ±82.5° regions toward poles.

To better understand these systematic differences, we investigate its latitudinal and altitudinal temporal variations over the
longitude sector 140°–160° E. This longitude sector was selected, because the subtropical jet stream seems to leave a quite
distinct feature over the western Pacific; differences of up to ±6 m s$^{-1}$ (i.e., up to exceeding the Table 1 threshold
requirements) are seen in this sector in the winter hemisphere (see Fig. 5a, b).
A 2007–2020 temporal analysis (Fig. 7) reveals that this pattern is systematic, with belts of positive and negative differences
within 10° to 40° latitude in the winter hemisphere (Fig. 7a, c). We therefore further define equally wide latitudinal belts that
encompass mentioned over- and underestimations to estimate the value of its temporal change. In January, we define the
regions of RO-derived wind over-/underestimation within the latitudinal belt 13°–23° N/28°–38° N, while for July it is 10°–
20° S/22°–32° S, respectively. The time-series analysis revealed that both positive and negative differences between the two
datasets decrease with time. Exceedance of the WMO-GCOS target requirement (Tab. 1) for long-term stability within ±0.5
m s$^{-1}$ per decade, taken as a consistency benchmark, is at 200 hPa level detected in both seasons for both positive and
negative differences (Fig. 7b, d). However, when calculated at other levels, e.g., 300 hPa or 250 hPa, change in the region of
RO-wind overestimation is in some seasons below ±0.5 m s$^{-1}$ threshold (Tab. 2).



The explanation to this result is shown in the Fig. 8. Besides the temporal change in its amplitude, we also find an altitudinal shift in the systematic difference. This is more expressed for belts where $RO_{geos}$ overestimates $ERA_{geos}$ wind. From that, we can conclude that an analysis of changes in the jet-stream core position based on ERA5 and RO data can give somewhat

different results. Furthermore, we find a clear noticeable decrease in the amplitude of the systematic difference after the year 2016 which coincides with a major observing system change in the ERA5 data assimilation (Hersbach et al., 2020; Fig. 3 and 4 therein).

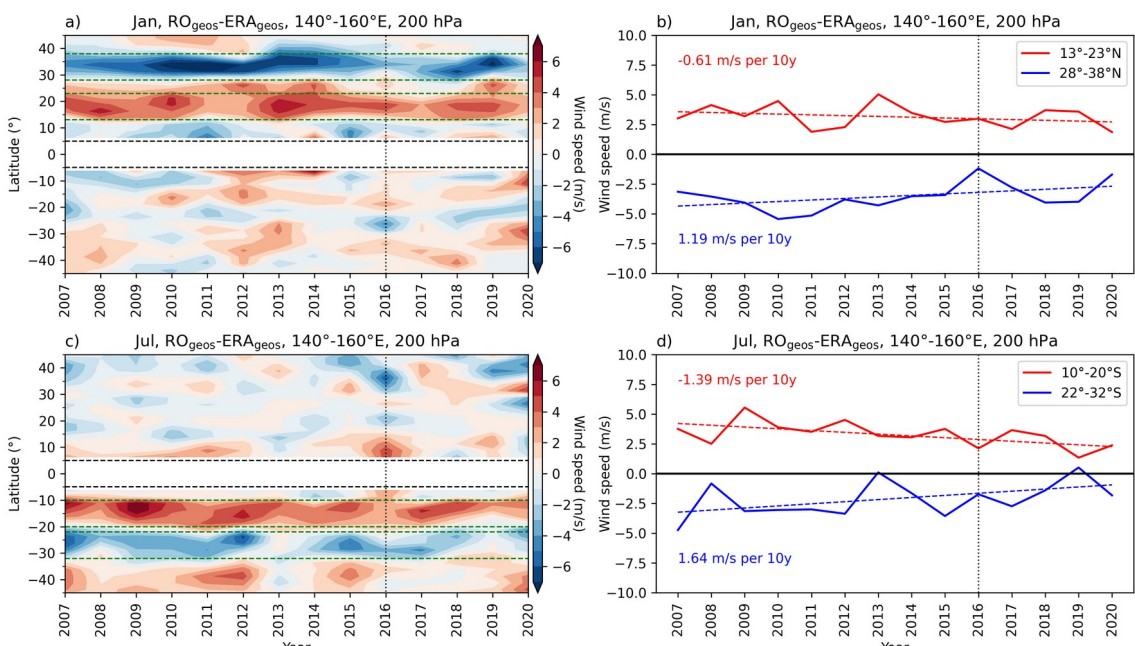

**Figure 7:** Latitudinal temporal distribution of the systematic difference between RO and ERA5. Wind speed difference (m s⁻¹) between the geostrophic RO ($RO_{geos}$) and the geostrophic ERA5 ($ERA_{geos}$) wind, averaged over the 140°–160° E area at 200 hPa (left column), for January (top) and July (bottom). Dashed green horizontal lines denote 10° latitudinal belts 13°–23° N and 28°–38° N in January, and 10°–20° S and 22°–32°S in July, for which the temporal trend analysis is made (right column and Table 2). Black dashed horizontal lines delineate the ±5° equator band and black dotted vertical lines mark the

year 2016, where ERA5 saw a change in observing systems.

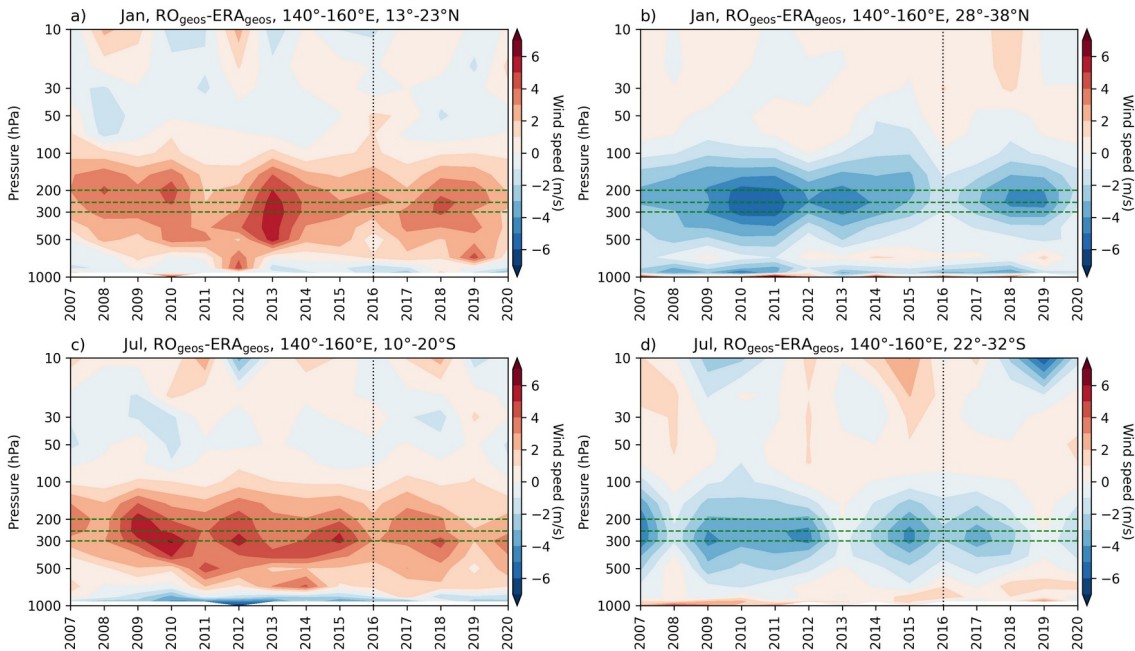

**Figure 8:** Altitudinal temporal distribution of the systematic difference between RO and ERA5. Wind speed difference (m s[-1]) between the geostrophic RO (RO$_{geos}$) and the geostrophic ERA5 (ERA$_{geos}$) wind, averaged over the 140°–160° E area for a) 13°–23° N and b) 28°–38° N in January, and c) 10°–20° S and d) 22°–32° S in July. Dashed green lines denote 200 hPa, 250 hPa and 300 hPa levels for which the decadal trend values are given in the Table 2. Black dotted vertical lines mark the year 2016, where ERA5 saw a change in observing systems.



**Table 2.** Decadal trend values (m s-1 per decade) in the systematic difference between the geostrophic RO and the geostrophic ERA5 wind speeds, averaged over 140°–160° E longitudinal area and latitudinal regions 13-23° N and 28°-38° N in January, and 10°-20° S and 22°-32° S in July, at 200 hPa, 250 hPa and 300 hPa levels. Systematic-difference trend rates larger than the WMO-GCOS (2016) long-term stability requirement of ±0.5 m s$^{-1}$ per decade (Table 1) are bold-faced.

| | | Trend rate (m/s per decade) | | |
|---|---|---|---|---|
| | | 200 hPa | 250 hPa | 300 hPa |
| January | 13-23° N | **-0.61** | -0.02 | -0.29 |
| | 28-38° N | **1.19** | **1.24** | **1.44** |
| July | 10-20° S | **-1.39** | **-1.07** | -0.47 |
| | 22-32° S | **1.64** | **1.61** | **1.53** |






### 3.3 Total bias – RO$_{geos/grad}$ vs. ERA$_{orig}$

In the final step, we show the total wind speed bias, i.e., how well is the original ERA5 wind field estimated by RO-derived
winds. Here, we evaluate the total wind speed bias in regard to WMO-related requirements (Tab. 1).

At the lowest observed level (200 hPa), in both seasons RO$_{geos}$ gives better estimates of ERA$_{orig}$ compared to RO$_{grad}$. For
RO$_{geos}$ these differences are generally below 2 m s$^{-1}$, except from the slight overestimation (~ 3 m s$^{-1}$) at ~15° winter
hemisphere latitude (Fig. 9a and b). Generally good agreement of RO$_{geos}$ and ERA$_{orig}$ is also observed at 150 hPa level, with a
bit lower overestimation at ~15° winter hemisphere latitude (Fig. 9c and d). At 50 hPa level, both approximations give
January wind estimations with bias bellow 2 m s$^{-1}$, while in July RO$_{geos}$ overestimates ERA$_{orig}$ wind at ~60° S for around 4 m
s$^{-1}$ (Fig. 9e and f). At the highest selected level (10 hPa), RO$_{geos}$ overestimates ERA$_{orig}$ for a bit more than 2 m s$^{-1}$ at 60° N,
while RO$_{grad}$ underestimates it for ~4 m s$^{-1}$ in high-latitude polar regions (Fig. 9g). On the other hand, in July, RO$_{geos}$
overestimation over southern hemisphere is larger than 10 m s$^{-1}$, while RO$_{grad}$ gives estimates with accuracy within 2 m s$^{-1}$
(Fig. 9h).
Hence, in the last step we show horizontal fields of the total bias between RO-derived and original ERA5 wind at selected
levels, where we show RO$_{geos}$ winds for the two lower levels, and RO$_{grad}$ wind in the upper two levels.

Based on the results shown in the previous sections, it can be concluded that in the lower levels the total bias illustrated in
Figure 10 a-d is mainly due to the systematic difference between the RO and ERA5 data. Hence, improving the
approximation method will not contribute significantly to a better wind speed estimation in these altitudinal regions. The
exception is the monsoonal region where probably neglected advection terms have a non-negligible contribution. It is also
noticeable that the region of the strongest jet-stream is generally well described with RO$_{geos}$ winds. On the other hand, the
total bias in the upper levels is mainly the result of the applied approximation (Fig. 10 e-h). Here, the high-latitude RO$_{grad}$
wind underestimation is caused by neglecting the horizontal advection terms, which are important during strong wave
activity.



**Figure 9:** Latitudinal distribution of the total bias. Wind speed difference (m s⁻¹) between the geostrophic RO (RO$_{geos}$) and the original ERA5 (ERA$_{orig}$) wind (blue line) and between the gradient RO (RO$_{grad}$) and the original ERA5 (ERA$_{orig}$) wind (red line), at the 200 hPa (first row), 150 hPa (second row), 50 hPa (third row) and 10 hPa (last row) level, for January (left






column) and July (right column). WMO-based accuracy requirement values are indicated in absolute terms at values of 2 m s$^{-1}$ (ABSrv2, dotted black line) and in relative terms at 5 %, with reference to the original ERA5 wind speed (RELrv5, dashed black line).



**Figure 10:** Long-term (2007–2020) mean total bias. Wind speed differences (m s$^{-1}$) between the geostrophic RO (RO$_{geos}$) and the original ERA5 (ERA$_{orig}$) wind, at 200 hPa (first row) and 150 hPa (second row) level, and between the gradient RO





(RO$_{grad}$) and the original ERA5 (ERA$_{orig}$) wind, at 50 hPa (third row) and 10 hPa (last row) levels for January (left column) and July (right column). Dashed black horizontal lines delineate the ±5° latitude band around the equator and ±82.5° regions
toward poles.

## 4 Discussion

Testing commonly used methods to estimate wind speed based on the thermodynamic data, showed that even though the gradient wind approximation is a generalization of the geostrophic balance, it does not imply that it will always give the
better estimation of the real wind. In accordance with earlier studies, the geostrophic approximation is a suitable method to describe a wind field in the troposphere (Boville 1987; Scherllin-Prischer et al., 2014; Verkhoglyadova et al., 2014). In this study, we also detected the effect of larger obstacles on the wind flow as a part of the ageostrophic component. These wave-like patterns are likely associated with orographic gravity waves (e.g., Smith, 1982). The characteristics of the gravity waves over large mountain ranges depend on the mountain dimensions (height, width and length) and its orientation in regard to the
wind flow, static stability profiles as well as on the mean wind speed (e.g., Holton, 2012).

Evidently, the combination of the large topographic obstacles and the dynamic jet-stream above it, results in quasi-stationary pressure/geopotential patterns above these mountains, which are also fingerprinted into the monthly-mean fields. Hence, by neglecting the vertical wind component, as well as advection terms, the geostrophic approximation is less accurately valid above larger mountains and obstacles. Depending on the resolution of the dataset used, this wave-like pattern might not be
seen in coarser spatial grids where such a pattern is averaged out. The geostrophic approximation was also not so successful in describing wind in regions of monsoonal circulation where larger underestimation is observed. By neglecting horizontal advective terms (as well as vertical motions), typical upper-air monsoonal circulation over these regions, the strongest at around 150 hPa (Trenberth et al., 2000), is not well caught.

On the other hand, a larger contribution of curvature effects, as well as very strong winds, contribute to a better stratospheric
wind field estimation by using the gradient wind approximation (Elson 1986; Boville 1987; Randel 1987). Significant geostrophic wind overestimation in these regions is reduced by retaining the centrifugal term in the equations of motion. The topographic effect is still noticeable in these altitudes, since orographic gravity waves are vertically propagating waves. Even though the SH winter polar jet-stream is well described using the balanced gradient winds, this is not the case at the NH where the winter polar jet-stream is more asymmetrical. Larger underestimation of the gradient wind compared to the
original wind is detected in high-latitude regions at ~80°N, which is also found in other papers (Elson 1986; Randel 1987). This is due to the effect of the Aleutian high in those regions, high pressure system commonly found in the NH stratosphere during the winter season (Colucci and Ehrmann, 2018; Elson 1986; Harvey and Hitchman, 1996).





Obviously, by neglecting horizontal advection terms (i.e., wave flux convergence terms) the effect of this pressure system on the jet-stream in this region (wave-stream interaction) is not well included resulting in the underestimation of the original 450 wind by the gradient wind balance. Elson (1986) tried to overcome this problem by solving momentum equations using perturbation method. He starts with the geostrophic approximation to estimate initial wind values which are then substituted back into the original momentum equations. Randel (1987) overcomes the same problem in the polar stratospheric regions by iteratively solving momentum equations with adding horizontal wave flux convergence terms, also starting with the geostrophic approximation. He compares derived winds to the winds obtained from the general circulation model and 455 comments that, although this approach gives a better estimation of the polar jet stream, it does not catch the subtropical jet stream as well as the geostrophic approximation does.

The analysis of the systematic differences between ERA5 and RO datasets, revealed generally a good agreement over the whole near-global troposphere-stratosphere domain. The exception are again areas above large mountains where the amplitude of the systematic difference between the datasets is the largest. Oscillatory patterns, arising from ERA5 data, were 460 also present in these differences. Even though the geopotential difference between the two data sets is quite small (here not shown), compared to the magnitude of the geopotential itself (below 1 %), we find that such small differences can lead to appreciable differences in wind speeds (more than 10 m s$^{-1}$), since these derive from the spatial derivatives of the isobaric geopotential fields.

One of the possible sources of these differences in geopotential may be due to the way these are constructed. In ERA5, 465 vertical pressure integration is performed bottom-up, resulting in vertical propagation of larger errors from the (lower) troposphere region. In contrast, RO-derived data work with downward vertical integration from the tenuous mesosphere, so that pressure integration is more robust and leading to smaller errors (Leroy, 1997; Scherllin-Pirscher et al., 2017). Hence, besides high vertical resolution, this is an advantage of RO, related to its "power of vertical geolocation" as described in detail by Scherllin-Pirscher et al. (2017). Another possible source of the absence of the patterns in RO monthly-means 470 resides in the relative orientation of wave phase surfaces to be detected and the line-of-sight of the radio-occultation rays between transmitter and receiver satellites (Hierro et al., 2018; Alexander et al., 2008, Fig. 2 therein). This relative geometry varies considerably and quite weakens the sensitivity of RO to the wave structures, then on top damped by the monthly-mean smoothing effect. Moreover, the ideal-gas (equation of state) and hydrostatic approximations, intrinsic in RO pressure profiles retrieval, are no longer valid in these wave-perturbed domains, contributing another basic smoothing effect (Steiner 475 and Kirchengast, 2000; Scherllin-Pirscher et al., 2017).

In summary these properties imply that RO limitations in high space-time resolution turn out as an advantage in more robust accuracy achieved in longer-term larger-scale averages as used in this study. Notwithstanding these properties, we emphasize at the same time that since the pioneering studies of Steiner and Kirchengast (2000) and Tsuda et al. (2000) many studies have proven the high value of RO data for gravity waves analyses (e.g., de la Torre and Alexander, 2005; de la Torre



et al., 2006; Hierro et al., 2018). However, these mainly focused on vertical wave propagation and the well-resolvable aspects, and on extracting wave anomalies rather than averaging them out.

Besides this wave-like patterns, differences up to ±6 m s⁻¹ are found in the sub-tropical jet-stream regions, while better agreement is detected at the upper levels. Systematic underestimation in sub-tropical jet-stream centre and overestimation in its equator-ward parts indicates a difference in jet-stream position between the two datasets. Mentioned deviations in the sub-

tropical jet-stream is analysed in detail by testing its long-term stability using long-term trend fits over 2007–2020. We estimated a trend magnitude of more than 0.5 m s⁻¹ per decade which exceeds WMO-GCOS (2016) long-term stability requirement. Besides temporal changes in the amplitude of the systematic difference, temporal differences in altitudinal direction are also observed.

A change in the bias between the two datasets is especially visible after the year 2016, where certain observing system

changes occurred in the observational input data assimilated into ERA5 (Hersbach et al., 2020). Specifically, a salient increase in the number of assimilated observations is seen around this year for surface pressure and specific humidity (Hersbach et al., 2020; Fig. 3 therein). In addition, the inclusion of WIGOS AMDAR (WMO Integrated Global Observing System, Aircraft Meteorological Data Relay) data in 2015 and the exclusion of some wind profilers and ACARS (Aircraft Communications Addressing and Reporting System) data in 2016 are likely further sources of the inhomogeneities

(Hersbach et al., 2020; Fig. 4 therein). This result indicates the potential advantage of RO-derived winds in terms of long-term stability for multi-decadal wind field monitoring, for example, to monitor the changes in large-scale circulation patterns such as the tropical-subtropical Hadley circulation (e.g., Weatherhead et al., 2018) or in the subtropical and polar jet streams, respectively.

This study advances on earlier initial studies to derive wind fields based on RO data (Scherllin-Pirscher et al., 2014, 2017;

Verkhoglyadova et al., 2014). By comparing original ERA5 wind and RO geostrophic wind, Scherllin-Prischer et al. (2014) commented that the difference is mainly caused by the wind approximation used, compared to the effect of RO retrieval errors. However, here we show that in the tropospheric region, the systematic difference between the datasets contributes more to the differences in the sub-tropical jet-stream region. Beyond this, we advanced on several essential aspects in this study. The finer horizontal resolution used here (2.5°) compared to those of previous studies (5°), allowed us to go more

equator-ward to reliably explore the region of the breakdown of the geostrophic approximation. While previous studies excluded tropical regions between ±10° or ±15° (based on the argument of Coriolis force becoming small), we found that it is reliably possible to only exclude the ±5° equatorial band.

In addition, compared to the earlier studies, where few specific years were selected for the initial analyses, here we analysed long-term wind speed means, and the decadal-scale temporal stability, which gave more robust results. The difference

between RO-derived winds and original winds from ERA5 was here clearly decomposed into one component from the wind approximation bias and another part coming from the systematic difference between the datasets.





## 5 Conclusions and perspectives

The main goal of this study was to test the general ability of RO-derived wind fields to represent original ERA5 winds. For this purpose, we decomposed the total wind speed bias into the contribution depending on the approximation method (approximation bias) and the contribution from the difference between the two datasets (systematic difference). First, the ability of conventionally used local force balance approximations, geostrophic and gradient wind balance, to represent original monthly-mean wind speeds was evaluated based on the ERA5 reanalysis data. The validation was made both horizontally (from equator up to ±82.5° and from 180° W to 180° E) and vertically (from the near-surface troposphere up to the middle stratosphere) over the global domain.

The successful performance of the methods is of great importance for enabling a reliable long-term dynamical wind field monitoring based on the thermodynamic mass field data, such as available in the form of RO-derived isobaric geopotential height data. Hence, in a second step, we tested how well RO-derived winds agree with the corresponding ones estimated from ERA5 reanalysis data. Here, we additionally test the temporal changes in this systematic difference to check for possible inhomogeneities. Finally, we show how well is the original wind field from ERA5 described by RO-derived wind fields.

Generally, the approximation bias and the systematic difference are both larger in the winter hemisphere when the atmosphere is more active and the wind speeds are stronger (Wu and Jehn 1972; Scherllin-Prischer et al., 2014, 2017; Verkhoglyadova et al., 2014). This is one of the reasons why early validation studies a few decades ago were mainly performed for the winter season (e.g., Elson 1986; Boville 1987; Randel 1987).

Main findings include:

- with the applied spatial resolution of 2.5° latitude × 2.5° longitude, it is possible to use the wind approximations equatorward as close as down to 5° latitude,
- it is justified to use the geostrophic approximation as a method to estimate winds in the troposphere, while for the stratospheric winds the additional inclusion of the centrifugal term through employing the gradient wind balance contributes to better wind estimation,
- in the troposphere, larger ageostrophic contributions are found over large mountain ranges and in the monsoon regions, due to neglecting the vertical wind component and/or advective terms in the equations of motion,
- orographic gravity wave effects in the monthly-mean geopotential height fields are found in ERA5 but not RO data, due to less susceptibility of RO to these smaller-scale perturbations,
- in the stratosphere, the largest bias of the gradient wind approximation is detected in polar regions, where the effect of wave-jet-stream interaction is not included due to neglecting horizontal advection terms in the equations of motion,





- the differences between RO and ERA5 geostrophic winds are generally small with values well within $\pm2$ m s$^{-1}$, except in the region of the sub-tropical jet-stream where patterns of latitudinal over- and underestimation are observed, pointing to possible differences in the jet-stream position between two datasets,

- trend analysis of the detected jet-stream difference showed an exceeding of the WMO-GCOS (2016) 0.5 m s$^{-1}$ per decade stability requirement, pointing to an inhomogeneity in ERA5 data due to observing system changes and potential added-value from the long-term stability of RO-derived wind field records,

- overall, the total difference between RO-derived wind (RO$_{geos}$ in the troposphere and RO$_{grad}$ in the stratosphere) and ERA5 original wind is small in monthly-mean wind fields, with differences in the troposphere mainly due to the systematic difference in datasets and in the stratosphere due to the approximation bias.

Despite this decent progress towards assessing the utility of RO records for wind monitoring, some problems and questions remain. One of the future goals is to create global wind-speed dataset based on the RO data. For these purposes the equatorial region $\pm5°$ latitude, which was excluded here, needs to be filled. Healy et al. (2020) showed that RO-derived zonal mean balance winds well quantify stratospheric zonal winds at the equator. However, this equatorial-balance-approximation approach did not provide information on geographically gridded wind fields and appears to lack information towards lower altitudes into the troposphere. Following Healy et al. (2020) and Scaife et al. (2000), Danzer et al. (2023) went a step further and derived wind fields in tropical region using the equatorial balance approximation. They also showed that the geostrophic approximation works well in estimating zonal mean zonal wind in this region, while larger deviations are observed for the meridional wind component.

Hence, it is needed to combine these RO-estimated wind fields based on three different methods (equatorial wind balance in the tropics, geostrophic wind in the troposphere and gradient wind in the stratosphere) to derive physically meaningful wind field dataset. Additional improvements in approximation methods are needed in the regions where advection terms showed to be important (e.g., monsoon region in troposphere and stratospheric near-polar region). For these purposes, we plan to employ and validate the methods proposed by Elson (1986) and Randel (1987). To overcome the problem of wind calculations over polar regions, we plan to test some of the extrapolation methods such as the one proposed by Elson (1986). Another avenue is the also-mentioned lower accuracy of RO data above about 30 km, mainly related to residual ionospheric biases. The potential of improving geopotential height data from RO at these altitudes does exist (e.g., Healy and Culverwell, 2015; Danzer et al., 2020, 2021; Liu et al., 2021; Syndergaard and Kirchengast, 2022), and the use of newest reprocessed RO data records is expected to also help improve wind monitoring in the upper stratosphere.

Overall, the added value of RO data is expected to be provided by its unique combination of high accuracy and long-term stability over inter-annual to decadal time periods of climate change relevance. This capacity to accurately keep long-term consistency valuably complements the dense resolution and coverage qualities of reanalyses, where occasional inhomogeneities due to changes in observing systems are experienced. The potential for climate-related studies is manifold,



and given the increasing observational database from the multi-satellite RO observing systems, RO data can serve as a valuable additional data source also for wind field monitoring.

**Author contribution**

Conceptualization: GK, JD; Data curation: IN; Formal analysis: IN, JD; Funding acquisition: JD; Investigation: IN, JD; Methodology: IN, JD, GK; Supervision: JD, GK; Validation & Visualisation: IN, JD, GK; Writing – original draft preparation: IN, JD; Writing – review & editing: IN, JD, GK.

**Acknowledgments**

We thank the UCAR/CDAAC RO team for providing RO excess phase and orbit data and the WEGC RO team for providing the OPSv5.6 retrieved profile data. We particularly thank F. Ladstädter (WEGC) for providing the monthly gridded climatology data and related discussions. Furthermore, we thank the ECMWF for providing access to the ERA5 reanalysis data. Finally, we thank the Austrian Science Fund (FWF) for funding the work; the wind analysis is part of the FWF stand-alone project Strato-Clim (grant number P-40182).


**Data Availability Statement**

The ERA5 data on pressure levels can be downloaded at (https://cds.climate.copernicus.eu/cdsapp#!/dataset/reanalysis-era5-pressure-levels?tab=form). The OPSv5.6 data are available at the website (https://www.doi.org/10.25364/WEGC/OPS5.6:2020.1).






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
