# Peer review of "The added value and potential of long-term radio occultation data for climatological wind field monitoring"

_Atmospheric Measurement Techniques, 2024_

## Referee Comment (RC1)

**Review for "The added value and potential of long-term radio occultation data for climatological wind field monitoring" by Nimac et al.**

The paper investigates wind fields derived from radio occultation (RO) data. Regions where geostrophic and gradient wind approximations are valid are identified by means of ERA5 data. RO derived wind fields are compared to ERA5 derived geostrophic and gradient winds and the full ERA5 wind fields.

The novelty of the study is that the analysis includes a comparison of geostrophic and gradient winds of ERA5 with the full winds in ERA5. Moreover, comparison of RO and ERA5 data are conducted on a higher resolved horizontal grid (2.5° x 2.5°) compared to earlier studies and lower latitudes close to the equator were also investigated. However, ERA5 analyses would be available on a 10 times higher resolved lat-lon grid (0.25° x 0.25°).

The paper is generally well structured, motivation and conclusion need to be strengthened, and some errors were found in the citations and references.

**Major Comment (Motivation, Conclusion, L7, L24):** The motivation and conclusion are in my view critical. The data product does provide no additional information to what is already available from ERA5. The accuracy was evaluated based on ERA5 which also has its limitations. But more importantly, ERA5 (and other reanalyses) are also available for long time periods that allow climate monitoring of wind fields up to mid-stratosphere. I think the important question is, can wind information determined from RO improve the wind fields in ERA5 reanalyses?

Aeolus did improve ECMWF IFS analyses and forecasts but as mentioned in the introduction, they were only available for a limited time period and therefore not feasible for reanalyses and climate time scales.
Additional independent evaluations with Aeolus winds and/or ECMWF IFS operational analyses (with Aeolus assimilated) (in addition to ERA5) would increase the robustness of the determined accuracy and the value of the additional data derived from RO but I agree that is probably beyond the scope of this paper. However, at probably a 2-months comparison of full resolution ECMWF IFS with Aeolus assimilated for one selected year could be added.

Additional comments:

Abstract:
L11: Please add here, that the focus is on synoptic scales (resolved by RO) because ERA5 native resolution is much higher.

L14/15: The third goal needs to be rephrased to be more specific: evaluate the potential of the RO wind fields for which purpose?

L17: Can you be more specific for what altitude range is meant by tropospheric? Presumably not in the surface boundary layer?

L19: For this latitude range and the exceptions that are given afterwards, I would not call it almost globally: ...within 2 m/s accuracy for latitudes 5°-82.5°, with some…

L20/L21: Please tell at the beginning of the abstract which altitude levels are investigated. What is considered as lower altitudes? Lower troposphere? What is considered as higher altitudes? Mid-stratosphere?

L21: very good → good; partly 4 m/s

L22ff: temporal change: is this a temporal increase? Differences between RO and ERA5 increase in time because ERA5 was potentially improved in 2016?

L56: good → please be more specific

L61: What kind of wave activity? Rossby waves? Gravity waves?

L72, general: What is new to what was already published by Scherllin-Pirscher et al. 2014? They investigated different latitudinal and altitudinal regions. They had 5°x5° lat-lon bins and also found 2 m/s with respect to full wind in ECMWF IFS analysis.

L81: See major comment one. ECMWF IFS analyses are an additional data set.

L104: depends somewhat also → remove somewhat

L112ff: This statement is not appropriate and cannot be made in general. The focus on wind the manuscript is totally different from von Schuckmann et al. who dealt with the atmospheric heat content. This needs to be proven, e.g. by adding additional (re)analysis data. See also major comment 1. Please check your references and bibliography carefully for correctness! Von Schuckmann et al. is 2020, here it says 2023, in the bibliography 2022. Moreover, it is in Earth Syst. Sci. Data and no longer as Discussion paper.

L117: Please clarify and motivate why a 2.5°x2.5° grid is used, the native resolution of ERA5 is higher. The differences between the approximations and the original wind fields are likely different for ERA5 data on the 0.25° x 0.25° grid. Please comment on this or show, that your main findings (e.g. the regions were the approximations break down) are robust. Otherwise make clear, that your findings are may not hold for higher resolution ERA5 (or ECMWF IFS analysis) data.

L131/140: accurately; high accuracy: Accuracy of the RO geopotential height is missing?

L133: Is the 2.5° x 2.5° the best what can be achieved from the RO data? This then motivates why ERA5 data is used at such low grid resolution and should be stated above.

L140ff: The altitude range must be mentioned in the abstract and in the introduction (see earlier comments above). Not just the upper level (10 hPa) but also the lowest level. In L97 is says 1000 hPa but here you say RO data are accurate above 5 km which is mid troposphere? Please clarify.

L151: bias → ageostrophic contributions/components.

L155/156: I don't understand what is meant by the sentence "This way…". How does the comparison mention in the sentence before tell anything with respect to the original winds?

L158: higher → upper troposphere

L161: I don't see how this is an evaluation of an additional value of the RO data. Better: To assess the temporal homogeneity and long-term stability of the RO data, we analyse the temporal evolution of the differences in wind derived from RO and ERA5.

L168ff: This is not an f-plane because f is not set constant but depends on latitude. Must be Holton and Hakim, 2012 and bibliography is wrong (An Introduction to Dynamic Meteorology, 4th edition, Academic Press is 2004). Eq. 2: acos → a cos

L189: remove "on top"

Eq 5: ageo(strophic) is a more appropriate term than bias

L200: Is this old reference of 1986 still valid for data of 2007-2020 used here? E.g., Scherllin-Pirscher et al. 2014 do not state anything about smoothing their fields (however they used a 5°x 5° lat-lon grid)

L242: The terms advantages or disadvantages is not appropriate here.

L246: Bias is not the right term in this context (see previous comments).

L252, Fig2-Fig6/Fig10: Because you only have a 2.5° latitude grid, the -5°/+5° band is essentially represented by 2 grid points along latitude. In order to make this clearly visible, the maps and vertical sections should be plotted pixelwise rather than using a contour plot with linear interpolation between grid points.

L255: delete obstacles. Gravity waves/mountain waves should me mentioned here. Greenland, the Andes, and the Himalaya are hotspots of mountain waves propagating into the stratosphere. In addition, the pattern at 60 S in July is a region of high gravity wave activity, known as the gravity wave belt.

L283: Did you also avoid the mountain regions on purpose? If this is the case, this should be mentioned here.

L285: Its not directly above Antarctica but rather at 60°S. (cf. comment above).

L291, Fig2-Fig4: It would help to also show ERA_grad-ERA_orig.

L356: Can you be more specific what those changes were (reference to Discussion section)? For July, a break similar break is found at 2008, 2013 and 2019. Without additional investigations, this statement seems rather speculative and I think it should be left out in the manuscript and 2016 not to be highlighted at all.

L399: systematic differences: here you need to be more specific; to what exactly are you referring to in the previous section? How can this be understood? Additional comparison with ECMWF IFS winds would help; they are not identical to ERA5 (see comments above).
better wind speed estimations: better with respect to what?

L426: But not in the Boundary layer.

L428: there are more recent (review) references to (orographic) GWs.

L527: specify more active? More mountain/gravity wave activity?

L591: ERA5 data from Copernicus are on 0.25° x 0.25° lat-lon grid. How did you get to 2.5° x 2.5° from there? Are you aware that 0.25° x 0.25° interpolated or reduced to a 2.5° x 2.5° lat-lon grid is not necessarily the same as directly downloading ERA5 data on the 2.5° x 2.5° lat-lon grid which depends on the configuration of the MARS request at ECMWF.

---

## Author Comment (AC1)

**Reviewer #1**

**Review for "The added value and potential of long-term radio occultation data for climatological wind field monitoring" by Nimac et al. – Author's Response to Reviewer #1**

The paper investigates wind fields derived from radio occultation (RO) data. Regions where geostrophic and gradient wind approximations are valid are identified by means of ERA5 data. RO derived wind fields are compared to ERA5 derived geostrophic and gradient winds and the full ERA5 wind fields.

The novelty of the study is that the analysis includes a comparison of geostrophic and gradient winds of ERA5 with the full winds in ERA5. Moreover, comparison of RO and ERA5 data are conducted on a higher resolved horizontal grid (2.5° x 2.5°) compared to earlier studies and lower latitudes close to the equator were also investigated. However, ERA5 analyses would be available on a 10 times higher resolved lat-lon grid (0.25° x 0.25°).

The paper is generally well structured, motivation and conclusion need to be strengthened, and some errors were found in the citations and references.

We thank the Reviewer #1 for the thorough review and constructive comments. We are now aware that we should have better highlighted that we used ERA5 data on a 2.5° resolution to test the quality compared to the RO wind-field retrieval, whose reasonable possible horizontal resolution is also 2.5°. Hence, we are only interested how well are the synoptic scale wind features described by the selected wind approximations. We also thank for the useful comment regarding direct download of 2.5° data, rather than post-processing 0.25° data which may introduce additional "residual noise" effects or other technically avoidable deviations. And indeed this has resulted in useful changes in that it helped to get rid of some residual "wiggle features" that we now could confirm had arisen from details of pre-processing at the smaller spatial scale. While it was hence beneficial to get rid of these and related more speculative considerations/discussions on GWs, all the general large-scale patterns discussed and the conclusions remained the same.

To better indicate the motivation and overall goal for this study, we modified the section in the Introduction (L84) as following:

*"Hence in this study the main goal is to develop an RO-based climatic wind data product over the free troposphere (troposphere region above planetary boundary layer - PBL) to the mid-stratosphere. The derived observations-based RO climatic wind fields have the potential to serve as a complementary climate-oriented dataset to reanalysis wind products. This is of specific interest, since the uncertainties and errors in reanalyses are less well understood and more complex, due to changes in the assimilated data, as well as uncertainties arising from the weather forecast model used and the assimilation method (Parker 2016; Hoffman et al., 2017). On the other hand, RO data are long-term stable, essentially free from satellite-to-satellite bias and hence requiring no inter-satellite calibration, which leads to better known uncertainties and clear error characteristics (Steiner et al., 2020).*

*The approach for the creation of RO climatic winds is three-fold. First, we test the approximation bias of the geostrophic and gradient-wind approximations. This serves as an information on the quality of this method for deriving monthly-mean winds based on the thermodynamic mass fields. For this purpose, we use ERA5 reanalysis data at a synoptic-scale spatial resolution which fits the horizontal resolution of RO (about 300 km, 2.5° × 2.5° grid). Second, we evaluate the difference between RO-derived winds and the ones estimated based on ERA5 data. Such a comparison of the systematic data bias helps reveal the added value of RO-derived monthly-mean winds as an independent wind field record. Lastly, we evaluate the potential of RO-estimated winds in representing "original" ERA5 wind fields. To this end, we compare RO long-term monthly mean winds with the actual winds in ERA5. To test the robustness of estimated RO climatic winds, we perform an additional comparison with the ECMWF-IFS operational analyses for two selected test months, in a time frame when Aeolus data were assimilated."*

**Major Comment (Motivation, Conclusion, L7, L24):**

The motivation and conclusion are in my view critical. The data product does provide no additional information to what is already available from ERA5. The accuracy was evaluated based on ERA5 which also has its limitations. But more importantly, ERA5 (and other reanalyses) are also available for long time periods that allow climate monitoring of wind fields up to mid-stratosphere. I think the important question is, can wind information determined from RO improve the wind fields in ERA5 reanalyses?

As commented above, we tried to make the motivation and main goal now clearer. We also adapted the discussion and conclusion parts to be more concise.

In this research we aim to develop climate-oriented RO wind fields, so we would not directly aim to show the potential of RO winds to improve wind fields in ERA5 reanalyses. We rather see it as a complementary and independent dataset to reanalysis data, for climate analysis and monitoring purposes. Regarding additional information that this data product offers, we see several benefits:

(i) As commented by Parker (2016), reanalysis uncertainties and errors are more complex and less well understood compared to those in observations or from well-understood satellite data retrievals. Besides, reanalyses can misinterpret certain phenomena due to spatially coherent and correlated errors (Hoffmann et al., 2017). Here, we perform a two-fold decomposition of the wind bias to better understand sources of differences. The first part tests the quality of the approximation itself (based on the ERA5), while the second part assesses the systematic difference between ERA5 and RO data.

(ii) Additionally, RO data are long-term stable, essentially free from satellite-to-satellite bias, highly precise and accurate, resulting in stable and well known uncertainty and error characteristics (Steiner et al., 2020). On the other hand, ERA5 reanalysis data meets changes in sources of assimilated data (Hersbach et al., 2020). Hence, one of our hypotheses was that temporal changes in systematic difference would point to possible inhomogeneities in ERA5 data. We hope to cover this topic in more details in our future studies.

(iii) Finer vertical resolution compared to the reanalysis data in the stratosphere region, together with improved RO data accuracy in those regions (ionosphere correction), might provide additional wind information in mid-stratosphere regions, where reanalyses can assimilate less data and the output is hence less observation-constrained and more strongly a result of the model.

We included these above-mentioned benefits in the conclusion part (L559 onwards).

Aeolus did improve ECMWF IFS analyses and forecasts but as mentioned in the introduction, they were only available for a limited time period and therefore not feasible for reanalyses and climate time scales. Additional independent evaluations with Aeolus winds and/or ECMWF IFS operational analyses (with Aeolus assimilated) (in addition to ERA5) would increase the robustness of the determined accuracy and the value of the additional data derived from RO but I agree that is probably beyond the scope of this paper. However, at probably a 2-months comparison of full resolution ECMWF IFS with Aeolus assimilated for one selected year could be added.

We appreciate this suggestion. We hence took it on board, despite it involved some significant additional effort, and included test-months comparisons with ECMWF-IFS operational analyses (with Aeolus assimilated), for February 2020 and July 2020. We added respective figures in the Appendix.

**Additional comments:**

Abstract:
L11: Please add here, that the focus is on synoptic scales (resolved by RO) because ERA5 native resolution is much higher.

Ok, added.

**L14/15: The third goal needs to be rephrased to be more specific: evaluate the potential of the RO wind fields for which purpose?**

We see the potential of RO-winds as a complementary dataset for wind analyses and wind monitoring, with added value due to its long-term stability and high-accuracy in the free troposphere and lower stratosphere. We rephrased the sentence (L15):
*"Third, by comparing the RO climatic winds to the ERA5 original winds, we evaluate the potential benefit of RO as an additional dataset for wind analyses and climate monitoring."*

**L17: Can you be more specific for what altitude range is meant by tropospheric? Presumably not in the surface boundary layer?**

Thank you for noticing this aspect where we had been unclear related to the altitudinal range we cover in our study. We refer to the free troposphere, i.e., we start from the 800 hPa level, which is above the planetary boundary layer (PBL). Apart from the inability of the selected approximations to describe the complex dynamics in the PBL, the RO-data quality in the PBL is also lower. Hence, RO-based climatic wind fields are created in the altitudinal range from 800 hPa to10 hPa. We added this clearly in the manuscript now (L109) and made it also more clear throughout the manuscript.

**L19: For this latitude range and the exceptions that are given afterwards, I would not call it almost globally: …within 2 m/s accuracy for latitudes 5°-82.5°, with some…**

Ok, accepted and corrected.

**L20/L21: Please tell at the beginning of the abstract which altitude levels are investigated. What is considered as lower altitudes? Lower troposphere? What is considered as higher altitudes? Mid-stratosphere?**

We agree we should have mentioned more clearly that we focus on the free troposphere to the mid-stratosphere region. We emphasized this more clearly throughout the manuscript.

**L21: very good → good; partly 4 m/s**

Ok, corrected.

**L22ff: temporal change: is this a temporal increase? Differences between RO and ERA5 increase in time because ERA5 was potentially improved in 2016?**

We choose here the term "temporal change", since positive (in underestimation) and negative (in overestimation) trends are observed pointing to a decrease in the difference between the two datasets.

**L56: good → please be more specific**

We added the following references: Holton and Hakim, 2013; Boville 1987; Randel 1987.

**L61: What kind of wave activity? Rossby waves? Gravity waves?**

Here it is referred to large amplitude planetary waves (i.e., Rossby waves).

**L72, general: What is new to what was already published by Scherllin-Pirscher et al. 2014? They investigated different latitudinal and altitudinal regions. They had 5°x5° lat-lon bins and also found 2 m/s with respect to full wind in ECMWF IFS analysis.**

Since our main motivation is the creation of a global RO-based wind field data product, we perform a systematic analysis which serves as a basis for this. There are in fact a range of advances in regard to

Scherllin-Pirscher et al. (2014), (SP2014). We use an improved horizontal resolution, which might result in more details and larger amplitudes of the total bias. We also more clearly divide the total bias into the contribution from the approximation itself, and the difference between datasets. This helps to better understand in which direction we should proceed: an advancement of the estimation methods and/or RO-data quality improvement. Unlike in SP2014, we perform a systematic analysis across the whole available time period up to 2020 and include regions close to the equator. SP2014 spared out 15°S-15°N, while we fully cover the tropics towards a small equatorial band, where a separate recent study by Danzer et al. 2024 provided a complementary analysis on also closing this gap across the equator. As another example on our improved data scope, SP2014 did not capture strong underestimation of polar winds by gradient RO winds, which is characteristic for years with a strong polar vortex. Besides, we try to link observed biases with dynamical processes and/or data quality issues in order to learn on this and account for it in our future work.

L81: See major comment one. ECMWF IFS analyses are an additional data

See at the beginning above; comment accepted and we added operational analyses together with climate models and reanalyses.

L104: depends somewhat also → remove somewhat

Ok, comment accepted and corrected.

L112ff: This statement is not appropriate and cannot be made in general. The focus on wind the manuscript is totally different from von Schuckmann et al. who dealt with the atmospheric heat content. This needs to be proven, e.g. by adding additional (re)analysis data. See also major comment 1. Please check your references and bibliography carefully for correctness! Von Schuckmann et al. is 2020, here it says 2023, in the bibliography 2022. Moreover, it is in Earth Syst. Sci. Data and no longer as Discussion paper.

We agree with your statement and we significantly rephrased the paragraph to make our formulation more careful (L131):
"*With respect to our first goal of the analysis, which is to test the quality of the two approximations, the specific selection of the reanalysis dataset would hardly make a difference in the obtained results. However, the systematic difference might change with a different reanalysis dataset. Considering the results from other studies that include also MERRA-2 and JRA-55 reanalyses (e.g., von Schuckmann et al. 2023, Sect. 3 therein, where atmospheric heat content change results built on changes in mass density fields), we can expect that the selection of the reanalysis dataset presumably has no major effect on the systematic difference for wind speeds derived from geopotential fields. We plan to perform a comparison with several reanalyses in our future research. In this study, as an additional evaluation of the robustness of the results, we utilize the ECMWF-IFS analysis data for February and July 2020, a period when Aeolus data were assimilated.*"

L117: Please clarify and motivate why a 2.5°x2.5° grid is used, the native resolution of ERA5 is higher. The differences between the approximations and the original wind fields are likely different for ERA5 data on the 0.25° x 0.25° grid. Please comment on this or show, that your main findings (e.g. the regions were the approximations break down) are robust. Otherwise make clear, that your findings are may not hold for higher resolution ERA5 (or ECMWF IFS analysis) data.

As stated above, the 2.5° x 2.5° grid was here selected since this is the possible horizontal resolution of the RO dataset. We are aware that the ageostrophic contribution for 0.25° ERA5 might be different (i.e., more expressed) compared to the 2.5° ERA5 data. However, investigating the effect of data resolution to robustness of the method is beyond the scope of this paper, since we focus here on the synoptics-scale and monthly-mean climate-oriented resolution.

L131/140: accurately; high accuracy: Accuracy of the RO geopotential height is missing?

We added two new references Scherllin-Pirscher et al. (2011a and 2011b) where further details about the accuracy of the geopotential height variable is given. The publications showed that the statistical error in the UTLS outside the tropics is on average smaller than 2 m, while the systematic error was estimated to be less than 7 m in the UTLS.

L133: Is the 2.5° x 2.5° the best what can be achieved from the RO data? This then motivates why ERA5 data is used at such low grid resolution and should be stated above.

Yes, 2.5° is the horizontal resolution of the RO data. As the first part of the analysis (evaluation of the methods) is made as a basis for the RO-wind retrievals, this is the reason why ERA5 data were also selected at 2.5°. We added this detail at several places in the manuscript.

L140ff: The altitude range must be mentioned in the abstract and in the introduction (see earlier comments above). Not just the upper level (10 hPa) but also the lowest level. In L97 is says 1000 hPa but here you say RO data are accurate above 5 km which is mid troposphere? Please clarify.

Following an earlier comment, we also added the lowest level and mentioned the altitudinal range of the derived data in the abstract, introduction section as well as data and study method parts.

L151: bias $\rightarrow$ ageostrophic contributions/components.

As we also examine here gradient wind (which represents a part of the ageostrophy), we label the bias as the approximation bias to cover both approximations simultaneously.

L155/156: I don't understand what is meant by the sentence "This way…". How does the comparison mention in the sentence before tell anything with respect to the original winds?

As the total difference is equal to the sum of the approximation bias and the systematic difference, this two-steps evaluation method helps to understand which bias contributes stronger to the total difference. We rephrased this sentence as:
"*The two-fold decomposition helps to attribute the individual contribution of each of the two biases (approximation and systematic) to the total difference between RO-derived wind field and ERA5 original winds.*" (L176)

L158: higher $\rightarrow$ upper troposphere

Ok, corrected.

L161: I don't see how this is an evaluation of an additional value of the RO data. Better: To assess the temporal homogeneity and long-term stability of the RO data, we analyse the temporal evolution of the differences in wind derived from RO and ERA5.

Our hypothesis is that due to the proven long-term stability of RO-data, detecting time changes in the systematic difference would point to inhomogeneities in ERA5 (the effect of changes in assimilation data). Our results, which also show an altitudinal change in the systematic bias, support the assumption that this might be the effect of changes in different sources of assimilated data in the reanalysis.

L168ff: This is not an f-plane because f is not set constant but depends on latitude. Must be Holton and Hakim, 2012 and bibliography is wrong (An Introduction to Dynamic Meteorology, 4th edition, Academic Press is 2004). Eq. 2: acos $\rightarrow$ a cos

Thanks for noticing, we edited this.

L189: remove "on top"

Ok, removed.

Eq 5: ageo(strophic) is a more appropriate term than bias

Since we analyze also the difference between original and gradient wind, using the phrase "ageostrophic term" would not be fully correct, since the gradient wind partially covers the ageostrophic contribution.

L200: Is this old reference of 1986 still valid for data of 2007-2020 used here? E.g., Scherllin-Pirscher et al. 2014 do not state anything about smoothing their fields (however they used a 5°x 5° lat-lon grid)

We initially performed the analysis using original RO geopotential and noticed a larger spatial variability (noise-like patterns) in the wind difference compared to the ERA5. After applying the 5-point horizontal smoothing, the RO-wind fields showed an improved, less noisy pattern. We removed the reference and edited the following sentence:
"*As the ERA5 geopotential field is derived by numerical integration, it is smoother compared to the observation-based RO geopotential field. Hence, we smoothed the 2.5° × 2.5° RO geopotential fields using a 5-point Gaussian filter in longitudinal and latitudinal direction.*" (L221)

L242: The terms advantages or disadvantages is not appropriate here.

Ok, we changed it to "strengths and weaknesses of geostrophic and gradient wind approximation".

L246: Bias is not the right term in this context (see previous comments).

We edited it to "ageostrophic contribution".

L252, Fig2-Fig6/Fig10: Because you only have a 2.5° latitude grid, the -5°/+5° band is essentially represented by 2 grid points along latitude. In order to make this clearly visible, the maps and vertical sections should be plotted pixelwise rather than using a contour plot with linear interpolation between grid points.

Ok, we accepted this comment and we modified the figures accordingly.

L255: delete obstacles. Gravity waves/mountain waves should me mentioned here. Greenland, the Andes, and the Himalaya are hotspots of mountain waves propagating into the stratosphere. In addition, the pattern at 60 S in July is a region of high gravity wave activity, known as the gravity wave belt.

We excluded most of the comments and discussions on gravity waves as such features were no longer present after using directly downloaded 2.5° ERA5 data (see also answer to the major comment at the beginning above).

L283: Did you also avoid the mountain regions on purpose? If this is the case, this should be mentioned here.

We did not avoid the mountains on purpose. We just selected this region since both, gradient wind in the upper troposphere and geostrophic wind in the mid-stratosphere, result in larger biases there.

L285: Its not directly above Antarctica but rather at 60°S. (cf. comment above).

We removed this sentence as we limit our vertical range to 800-10 hPa.

L291, Fig2-Fig4: It would help to also show ERA_grad-ERA_orig.

We initially had such a display, but since for the larger part the patterns were similar to those between geostrophic ERA and original ERA, we decided that by showing the absolute difference between the two biases points to the most important details.

L356: Can you be more specific what those changes were (reference to Discussion section)? For July, a break similar break is found at 2008, 2013 and 2019. Without additional investigations, this statement seems rather speculative and I think it should be left out in the manuscript and 2016 not to be highlighted at all.

We hypothesized that such an approach would reveal inhomogeneities in ERA5, since for RO data long-term stability was proven (Steiner et al., 2020). We accept your comment and do not highlight the year 2016 specifically. We just indicate that there are temporal variations in the systematic difference between the two datasets.

L399: systematic differences: here you need to be more specific; to what exactly are you referring to in the previous section? How can this be understood? Additional comparison with ECMWF IFS winds would help; they are not identical to ERA5 (see comments above).
better wind speed estimations: better with respect to what?

Thank you for noticing this. This conclusion was not following the results previously shown, so we edited the paragraph:
"*Based on the results shown in the previous sections, it is clear that the total bias in the sub-tropical jet stream is a result of the systematic difference and the approximation bias (Fig. 10a-b). On the other hand, a larger underestimation of winds by ROgeos in the monsoonal region, as well as dipole structures related to stationary waves are mainly a result of the inability of the geostrophic approximation to capture such a circulation (Fig. c-d). Similar patterns can be seen in the Figure A1a and b, where a comparison between RO geostrophic winds and ECMWF-IFS analysis winds at 200 hPa level for February and July 2020 is given. However, the Figure A1 shows more details and a larger spatial variability compared to Fig. 10, where variability is lower due to temporal averaging.*
*In the upper levels, the total bias is mainly the result of the applied approximation (Fig. 10e-h). The exception is the tropical region where lower RO data quality contributes to somewhat larger bias. The NH high-latitude ROgrad wind underestimation in January is caused by neglecting the horizontal advection terms, which are important during condition when strong wave activity and polar night jet-stream interact. Again, the robustness of the results is supported by a further comparison with ECMWF-IFS analysis data (Appendix Fig. A1c and d). Similar patterns are observed with stronger expressed noise-like differences near the equator.*" (L405)

L426: But not in the Boundary layer.

Ok, we added "free" before "troposphere".

L428: there are more recent (review) references to (orographic) GWs.

As mentioned earlier above, we excluded most of the comments on GWs in the manuscript.

L527: specify more active? More mountain/gravity wave activity?

With the term "more active" we refer to a higher dynamic in terms of larger wind amplitudes and variability, as well as more wave activity. We have rewritten the sentence to:
"*Generally, the approximation bias and the systematic difference are both larger in the winter hemisphere when the atmosphere is more dynamic in terms of larger wind speeds and stronger wave activity*". (L499)

L591: ERA5 data from Copernicus are on 0.25° x 0.25° lat-lon grid. How did you get to 2.5° x 2.5° from there? Are you aware that 0.25° x 0.25° interpolated or reduced to a 2.5° x 2.5° lat-lon grid is not necessarily the same as directly downloading ERA5 data on the 2.5° x 2.5° lat-lon grid which depends on the configuration of the MARS request at ECMWF.

Thank you very much again for this valuable comment. We downloaded the 2.5°x2.5° ERA5 data via MARS request at ECMWF and re-plotted all the figures. The wave-like patterns over the large mountain areas essentially vanished, so we exchanged all the relevant figures with the new data and adapted the text accordingly. We highly appreciate your comment which helped us safe speculative interpretations that now could be explained by the details of different dataset pre-processing.

**Reference list (citations in this Response to Reviewer):**

Hersbach, H., Bell, B., Berrisford, P., Hirahara, S., Horányi, A., Muñoz-Sabater, J., ... and Thépaut, J. N.: The ERA5 global reanalysis, *Q. J. R. Meteorol. Soc.*, 146(730), 1999–2049, https://doi.org/10.1002/qj.3803, 2020.

Hoffman, R. N., Privé, N., and Bourassa, M. Comments on "Reanalyses and observations: What's the difference?". Bulletin of the American Meteorological Society, 98(11), 2455-2459. https://doi.org/10.1175/BAMS-D-17-0008.1, 2017.

Parker, W. S. Reanalyses and observations: What's the difference?. Bulletin of the American Meteorological Society, 97(9), 1565-1572. https://doi.org/10.1175/BAMS-D-14-00226.1, 2016.

Scherllin-Pirscher, B., Steiner, A. K., Kirchengast, G., Kuo, Y.-H., and Foelsche, U.: Empirical analysis and modeling of errors of atmospheric profiles from GPS radio occultation, Atmos. Meas. Tech., 4, 1875–1890, https://doi.org/10.5194/amt-4-1875-2011, 2011a.

Scherllin-Pirscher, B., Kirchengast, G., Steiner, A. K., Kuo, Y.-H., and Foelsche, U.: Quantifying uncertainty in climatological fields from GPS radio occultation: an empirical-analytical error model, Atmos. Meas. Tech., 4, 2019–2034, https://doi.org/10.5194/amt-4-2019-2011, 2011b.

von Schuckmann, K., Minière, A., Gues, F., Cuesta-Valero, F. J., Kirchengast, G., Adusumilli, S., Straneo, F., Ablain, M., Allan, R. P., Barker, P. M., Beltrami, H., Blazquez, A., Boyer, T., Cheng, L., Church, J., Desbruyeres, D., Dolman, H., Domingues, C. M., García-García, A., Giglio, D., Gilson, J. E., Gorfer, M., Haimberger, L., Hakuba, M. Z., Hendricks, S., Hosoda, S., Johnson, G. C., Killick, R., King, B., Kolodziejczyk, N., Korosov, A., Krinner, G., Kuusela, M., Landerer, F. W., Langer, M., Lavergne, T., Lawrence, I., Li, Y., Lyman, J., Marti, F., Marzeion, B., Mayer, M., MacDougall, A. H., McDougall, T., Monselesan, D. P., Nitzbon, J., Otosaka, I., Peng, J., Purkey, S., Roemmich, D., Sato, K., Sato, K., Savita, A., Schweiger, A., Shepherd, A., Seneviratne, S. I., Simons, L., Slater, D. A., Slater, T., Steiner, A. K., Suga, T., Szekely, T., Thiery, W., Timmermans, M.-L., Vanderkelen, I., Wjiffels, S. E., Wu, T., and Zemp, M.: Heat stored in the Earth system 1960–2020: where does the energy go?, Earth Syst. Sci. Data, 15, 1675–1709, https://doi.org/10.5194/essd-15-1675-2023, 2023.

Steiner, A. K., Ladstädter, F., Ao, C. O., Gleisner, H., Ho, S.-P., Hunt, D., Schmidt, T., Foelsche, U., Kirchengast, G., Kuo, Y.-H., Lauritsen, K. B., Mannucci, A. J., Nielsen, J. K., Schreiner, W., Schwärz, M., Sokolovskiy, S., Syndergaard, S., and Wickert, J.: Consistency and structural uncertainty of multi-mission GPS radio occultation records, Atmos. Meas. Tech., 13, 2547–2575, https://doi.org/10.5194/amt-13-2547-2020, 2020.

---

## Author Comment (AC2)

**Reviewer #2**
**Review of Nimac et al.: The added value and potential of long-term radio occultation data for climatological wind field monitoring", submitted to Atmospheric Measurement Techniques. – Author's Response to Reviewer #2**

The paper reports on a study of global wind-field monitoring from the middle troposphere to the middle stratosphere using the satellite-based Radio Occultation (RO) technique. The focus is on climate applications, hence monthly means and a rather coarse 2.5x2.5 degree latitude-longitude resolution. The latter is also a consequence of the limitations of the RO observing system.

Winds are derived from the gridded RO data by means of the geostrophic and gradient wind approximations. The idea is to use ERA5 reanalysis data as a reference in two ways: by computing wind fields from ERA5 using the same approximation as for RO, and by using the original ERA5 winds. In this way the approximation itself can be evaluated (ERA approximation vs original), RO data can be evaluated (RO vs ERA approximation), and the general ability of RO-derived wind fields to represent the real winds can be evaluated (RO vs ERA original). Key questions addressed are where and under what circumstances the approximation holds and RO can be expected to provide useful information on the wind.

Scientifically, the study presented is perhaps not a major leap forward, but it is a well-designed study, it fills a gap in the literature, it is well-written and easy to follow and it provides practically useful information. For anyone interested in generating atmospheric wind fields from RO data, or that is interested in the validity of the geostrophic and gradient wind approximations, this paper will provide highly valuable information.

The manuscript is well worth to be published. I mainly have one set of questions that I would like to see clarified. That relates to sampling and data retrievals. See the comments and questions below. I believe that addressing these questions and issues will improve the manuscript.

We are thankful to Reviewer #2 for the valuable comments regarding details on the data retrievals. We added more information about it in the manuscript.

**Comments and questions**

In Section 2.2, it is described how the monthly 2.5x2.5 degree grids are derived for the RO satellite data. For a calendar month, all RO profiles within 600 km from the center of a grid box are averaged with data weighted less with increasing distance from the center, following a Gaussian.
1) What is the width of the Gaussian?

We now explained this part better in the manuscript and added:
*"The monthly-mean fields are calculated based on the daily RO climatological fields, which are created by temporal and spatial weighting of RO atmospheric profiles. Temporal weighting is carried out within ±2 days, while spatial weighting is done within the constant radius of 600 km in order to maintain effective horizontal resolution. The profiles are weighted based on their distance from the center location of a bin with a bivariate (latitude–longitude) Gaussian function, which has a peak at the center of the bin and corresponding standard deviation of 150 km in latitudinal and 300 km in longitudinal direction, respectively. Details are given in the presentation by Ladstädter (2022)."* (L160)

2) Doesn't this mean that there are more data points per monthly mean at high latitudes compared to low latitudes, given the polar orbits of the satellites? Does this have any implications for the sampling errors of the monthly means?

Yes, however this is compensated for by larger variability in these regions (Scherllin-Pirscher et al., 2011).

3) Are the monthly means adjusted for sampling errors? It appears to be used in other studies using RO satellite data for climate studies.

Yes, the data we used are sampling-error corrected. We added this information in the manuscript (L166).

RO data retrievals: The key variable used in the study is the geopotential height of isobaric surfaces, i.e. geopotential height as a function of pressure. How is pressure retrieved? It is mentioned that "background information (re)analys data" is used in the retrieval. As an alternative, the pressure can be retrieved from the refractivity without such background information ("dry" retrieval).
4) Which data are used as background? If ECMWF reanalysis is used, does it have any consequences for the comparison between RO and ERA5?

The background data (short-range forecasts) from ECMWF-IFS/ERA5 are used in the regions, where dry variables differ much from the physical (actual) ones. This is the case in the moist lower troposphere where physical parameters are derived using the moist-air retrieval algorithm by combining individual profiles and background information (Li et al., 2019). In the UTLS region, the information is purely derived from the RO data. The variables dry density, pressure and temperature are estimated based on the refractivity equation, the downward integration of the hydrostatic equation and the equation of state. We provide a clarification for the selection of ERA5 reanalysis as a reference dataset despite its "RO dependence" in the "Data and study method" section (L123).

5) Is the "dry" retrieval used in this study, e.g., at altitudes where humidity is small? If so, describe this.

Yes, we use the dry retrieval, as discussed in the previous comment.

In Section 2.1, it is briefly described that the geostrophic and gradient winds are computed from isobaric geopotential height data on a monthly 2.5x2.5 degree grid.
6) To avoid that sampling errors have an impact on the results, wouldn't it be an idea to extract ERA5 profiles co-located with the RO profiles and then compute the ERA5 wind fields from that sampled data set? Or is that the way it has been done? If not, could you comment on that, e.g., why sampling issues are not likely to change the conclusions of the study.

See answer earlier above; we use sampling-error corrected RO data and hence the sampling error is not likely to change the conclusions of the study (for details on sampling error estimation see, e.g., Scherllin-Pirscher et al., 2011, 2017).

You also mention that you apply a 5-point Gaussian filter to the geopotential fields before computing the geostrophic winds.
7) What is the reason for this additional smoothing? Do you apply this smoothing to both RO and ERA5, or only to RO?

The smoothing is applied only to the RO data due to a larger spatial variability (i.e., noise features) observed in the initially estimated winds based on the non-smoothed geopotential. As ERA5 is based on a numerical integration, which produces much smoother fields compared to the observation-based RO product, it was necessary to smooth RO geopotential field.

**Reference list (citations in this Response to Reviewer):**

Ladstädter, F.: Talk on gridding strategies, in: OPAC-IROWG 2022 conference, Seggau, Austria, Seggau Castle, 8–14 September 2022, https://static.uni-graz.at/fileadmin/veranstaltungen/opacirowg2022/programme/08.9.22/AM/Session_1/OPAC-IROWG-2022_Ladstaedter.pdf (last access: 27 September 2024), 2022.

Scherllin-Pirscher, B., Kirchengast, G., Steiner, A. K., Kuo, Y.-H., and Foelsche, U.: Quantifying uncertainty in climatological fields from GPS radio occultation: an empirical-analytical error model, Atmos. Meas. Tech., 4, 2019–2034, https://doi.org/10.5194/amt-4-2019-2011, 2011.

Scherllin-Pirscher, B., Steiner, A. K., Kirchengast, G., Schwärz, M., and Leroy, S. S.: The power of vertical

geolocation of atmospheric profiles from GNSS radio occultation, J. Geophys. Res. Atmos., 122, 1595–1616, https://doi.org/10.1002/2016JD025902, 2017.

Li, Y., Kirchengast, G., Scherllin-Pirscher, B., Schwaerz, M., Nielsen, J. K., Ho, S. P., and Yuan, Y. B.: A new algorithm for the retrieval of atmospheric profiles from GNSS radio occultation data in moist air and comparison to 1DVar retrievals, Remote Sens., 11(23), 2729, https://doi.org/10.3390/rs11232729, 2019.

---

## Author Response (AR2)

**Public justification (visible to the public if the article is accepted and published):**

The authors have in my view well addressed the review comments and provided key clarifications to the manuscript. One point remains blurred, which is on spatial and temporal representation.

The authors comment that the physical limitation of RO data in terms of horizontal resolution is about 300 km, hence the RO data set cannot have a better spatial resolution, though spatial sampling could be anything. According to the definitions I know, sampling does not improve resolution, but may help to better represent a field (i.e., Nyquist sampling). Moreover, the GO data set is further blurred by a 2-day temporal window. How much is the spatial resolution decreased by such broad window, in which the wind has moved the air by about 3500 km (at 20 m/s) ? Finally, a 150/300 km SD Gaussian filter blurs the RO signal furthermore, again reducing the spatial resolution of the resulting GO fields. Admittedly, the latter step also reduces the GO noise and is thus useful for analyzing the remaining blurred signal.

On the IFS and ERA5 side, the 2-day filter may be missing, while model winds are much filtered due to spatial and temporal diffusion operators. This appears most clearly in spatial analyses of collocated wind observations and model data.

It is clear that the ERA5 and RO data have different spatiotemporal representation. The manuscript would much improve when the distinction between sampling and resolution was made and unjustified claims on 2.5 degree resolution were removed.

Furthermore, a better estimate of resolution would obviously benefit the manuscript as contributions of the geostrophic and ageostrophic components of the atmospheric flow will depend on the resolution (rather than sampling) of the instrument.

We thank to the Editor for drawing our attention to this aspect where we misuse the term "resolution". We corrected "2.5° resolution" claims to "2.5° spatial grid" and added few sentences regarding the differences between ERA5 and RO spatio-temporal representation in the Study method section (L225).

*"Such filtering smoothes not only the noise component, but part of the signal too. Hence the spatial resolution of the field decreases (Vishwakarma et al., 2018). The amplitude of the damped signal depends on the selected type of filter. For a Gaussian filter the resulting resolution of the fields gets coarser by a factor of two, compared to the smoothing radius (Devarju, 2015). Overall, it is clear that there are differences in the spatio-temporal representation of ERA5 and RO data. Regarding the temporal component, we can assume that temporal weighting applied to the daily RO profiles does not strongly influence the monthly-mean value. On the other hand, ERA5 (and ECMWF-IFS model) winds are also filtered by spatial and temporal diffusion operators (Hersbach et al., 2020). In summary, these post-processing methods affect the physical spatial resolution of the field. To estimate the effective physical resolution of the resulting climatic field is hence not an easy task (e.g., Vishwakarma et al., 2018). We plan to investigate this aspect more thoroughly in our future research by testing various filtering options and inspecting their influence on wind fields over mountainous regions, where fine horizontal structures are usually observed."*